# The mini-IDLE 3D biomimetic culture assay enables interrogation of mechanisms governing muscle stem cell quiescence and niche repopulation

Erik Jacques[1,2], Yinni Kuang[2,3], Allison P Kann[4,5,6], Fabien Le Grand[7], Robert S Krauss[4,5,6], Penney M Gilbert[1,2,3]*

[1]Institute of Biomedical Engineering, University of Toronto, Toronto, Canada; [2]Donnelly Centre, University of Toronto, Toronto, Canada; [3]Department of Cell and Systems Biology, University of Toronto, Toronto, Canada; [4]Department of Cell, Developmental, and Regenerative Biology, Icahn School of Medicine at Mount Sinai, New York, United States; [5]Black Family Stem Cell Institute, Icahn School of Medicine at Mount Sinai, New York, United States; [6]Graduate School of Biomedical Sciences, Icahn School of Medicine at Mount Sinai, New York, United States; [7]Université Claude Bernard Lyon 1, CNRS UMR 5261, INSERM U1315, Institut NeuroMyoGène - Pathophysiology and Genetics of Neuron and Muscle, Lyon, France

*For correspondence:
Penney.Gilbert@utoronto.ca

Competing interest: The authors declare that no competing interests exist.

**Abstract** Adult skeletal muscle harbours a population of muscle stem cells (MuSCs) that are required for repair after tissue injury. In youth, MuSCs return to a reversible state of cell-cycle arrest termed 'quiescence' after injury resolution. Conversely, some MuSCs in aged muscle remain semi-activated, causing a premature response to injuries that results in incomplete repair and eventual stem cell depletion. Regulating this balance between MuSC quiescence and activation may hold the key to restoring tissue homeostasis with age, but is incompletely understood. To fill this gap, we developed a simple and tractable in vitro method, to rapidly inactivate MuSCs freshly isolated from young murine skeletal muscle, and return them to a quiescent-like state for at least 1-week, which we name mini-IDLE (**I**nactivation and **D**ormancy **LE**veraged in vitro). This was achieved by introducing MuSCs into a 3D bioartificial niche comprised of a thin sheet of mouse myotubes, which we demonstrate provides the minimal cues necessary to induce quiescence. With different starting numbers of MuSCs, the assay revealed cellular heterogeneity and population-level adaptations that converged on a common niche repopulation density; behaviours previously observed only in vivo. Quiescence-associated hallmarks included a Pax7+CalcR+DDX6+MyoD-c-FOS- signature, quiescent-like morphologies, and polarized niche markers. Leveraging high-content bioimaging pipelines, we demonstrate a relationship between morphology and cell fate signatures for possible real-time morphology-based screening. When using MuSCs from aged muscle, they displayed aberrant proliferative activities and delayed inactivation kinetics, among other quiescence-associated defects that we show are partially rescued by wortmannin treatment. Thus, the assay offers an unprecedented opportunity to systematically investigate long-standing queries in areas such as regulation of pool size and functional heterogeneity within the MuSC population, and to uncover quiescence regulators in youth and age.

## Editor's evaluation

This methods paper explores procedures evaluating the balance between muscle cell quiescence and activation. These could well permit investigations of long-standing questions in key areas of muscle function. The latter include the regulation of adult stem cell pool size and functional heterogeneities in this, as well as regulators of muscle quiescence.

## Introduction

Muscle stem cells (MuSCs) are an adult stem cell population identifiable by the selective expression of the paired-box transcription factor Pax7 in skeletal muscle tissue, and are essential to muscle regeneration (*Lepper et al., 2011*; *Murphy et al., 2011*; *Sambasivan et al., 2011*; *Günther et al., 2013*; *von Maltzahn et al., 2013*). At rest, MuSCs exist in a reversible state of quiescence characterized by, among other features, the absence of cell-cycle indicators (*Chakkalakal et al., 2014*; *Cutler et al., 2022*), lowered metabolic activity (*Rocheteau et al., 2012*) and RNA content (*Machado et al., 2017*), increased expression of genes such as CalcR, CD34, Spry-1, and Sdc4 (*Machado et al., 2017*; *Pillon et al., 2020*; *Quarta et al., 2016*), and revealed more recently – an elaborated morphology (*Verma et al., 2018*; *Kann et al., 2022*; *Ma et al., 2022*). Anatomically, they reside between a myofiber and the surrounding basal lamina, a highly specialized microenvironment or 'niche', that conveys unto them the popularized term 'satellite cell' (*Mauro, 1961*). Though quiescent, they are not dormant but are in fact idling; constantly communicating with their niche and waiting to respond to stressors (*Van Velthoven and Rando, 2019*; *Crist et al., 2012*). Examples include significant physical activity causing mechanically induced damage, acute trauma, or exposure to myotoxic compounds that induce myofiber degradation (*Baghdadi and Tajbakhsh, 2018*; *Murach et al., 2021*). In these situations, MuSCs rapidly shift to an activated state wherein they enter cell cycle, and proliferate to produce progeny that differentiate to repair or create new myofibers, or they undertake self-renewing divisions where a subpopulation eventually returns to quiescence and repopulates the niche (*Rudnicki et al., 2008*).

Quiescence is required for the long-term stability of the stem cell pool, and proper activation kinetics are necessary to ensure the integrity of the repair process (*Ancel et al., 2021*). MuSCs are regarded as existing individually along a quiescence activation spectrum where shifts occur during different stages of regeneration (*Ancel et al., 2021*). The depth of quiescence is shown to be positively correlated with stem cell potency, or 'stemness'. Indeed, instances where depth of quiescence is lost, such as in aging, leads to less efficient and incomplete regeneration, and a progressive decline in MuSC number (*Chakkalakal et al., 2012*). However, how the process of MuSC inactivation (reversibly rendering non-active or inert) and the quiescent state are regulated, in youth and in age, remains largely unexplored, in part due to a reliance on in vivo studies, which imparts through-put limitations.

A requirement of quiescence studies in vitro (a setting that offers higher-content and precise experimental control as compared to in vivo studies), is that MuSC activation must be overridden to reinstate a quiescent state. This is because tissue dissection, enzymatic digestion, and cell sorting impart an injury-associated stress response to MuSCs, and cause isolation-induced activation (*Machado et al., 2021*). Several studies describe in vitro approaches to delay activation for days (at most) by manipulating the culture substrate or media (*Charville et al., 2015*; *Arjona et al., 2022*; *Monge et al., 2017*). Though temporary, these strategies offer potential options to be able to first isolate and expand MuSC number ex vivo and then preserve or improve MuSC potency by holding them in a quiescent state prior to transplantation (*Charville et al., 2015*; *Arjona et al., 2022*; *Monge et al., 2017*). Alternatively, combining a chemically defined 'quiescence media' with artificial muscle fibers was reported to maintain MuSCs in culture with limited proliferative activity or changes to cell volume, and sustained CD34 expression for a 3.5-day period (*Quarta et al., 2016*). More recently, three-dimensional (3D) skeletal muscle macrotissue platforms were shown to support Pax7[+] reserve cells (*Baroffio et al., 1996*, *Yoshida et al., 1998*) within human myoblast populations to take on a reversible quiescent-like state (*Tiburcy et al., 2019*; *Juhas et al., 2018*; *Fleming et al., 2020*; *Rajabian et al., 2021*; *Trevisan et al., 2019*; *Wang et al., 2022a*). To date, a strategy to inactivate freshly isolated adult MuSCs in culture for >3.5 days and to induce multiple molecular and morphological hallmarks of quiescence has yet to be reported. These experimental roadblocks, in turn, have focused all evaluations of MuSCs isolated from aged tissue on proliferation since studies of functional defects associated with quiescence have been precluded ex vivo.

**eLife digest** When our muscles are injured, stem cells in the tissue are activated to start the repair process. However, when there is no damage, these cells tend to stay in a protective, dormant state known as quiescence. If quiescence is not maintained, the stem cells cannot properly repair when the muscle is damaged. This happens in old age, when a proportion of the cells remain semi-activated, and become depleted. However, researchers still do not fully understand how quiescence is regulated. This is partly because in order to study quiescence, live animals must be used, because muscle stem cells immediately come out of quiescence when they are removed from muscle tissue.

To overcome this experimental limitation, Jacques et al. developed a new method to study muscle stem cells by transferring them from mice into three-dimensional engineered muscle tissue grown in the lab. This tissue is made by infiltrating the pores of teabag paper with muscle progenitor cells, which then fuse with one another to make a thin muscle that contains three layers of contractile muscle cells. Introducing muscle stem cells from young healthy animals into this engineered muscle tissue allowed them to return to a quiescent-like state and to remain in that state for at least a week. Cells from older animals could also be returned to dormancy if they were chemically treated after placing them in the engineered muscle tissue.

The approach works in a miniaturized fashion, with each engineered tissue requiring less than one per cent of the muscle stem cells collected from each mouse. This allows 100 times as many experiments compared to the current methods using live animals.

This system could help researchers to study the genetic and chemical influences on muscle stem cell quiescence. Further understanding in this area could lead to treatments that restore healing abilities in older muscle tissue.

To address these gaps and offer unparalleled access to the elusive quiescent MuSC, our group employed tissue engineering principles to create a scalable in vitro biomimetic niche capable of provoking inactivation in freshly isolated MuSCs. Previously, we invented MEndR, a method to study skeletal muscle endogenous repair 'in a dish' in a 24-well format by introducing MuSCs into thin sheets of engineered muscle tissue that we then injured using myotoxins (*Davoudi et al., 2022*). In our uninjured control tissues, we observed a non-negligible proportion of the engrafted cells remained mononucleated at the assay endpoint, in spite of the differentiation-inducing culture media used. We hypothesized that the muscle tissues were providing a pro-quiescence niche. To evaluate this in depth, herein we produced miniaturized (96-well format) muscle tissues, derived from primary mouse myoblasts, into which we introduced freshly sorted adult mouse MuSCs. We report that within these biomimetic niches, termed mini-IDLE (**I**nactivation and **D**ormancy **LE**veraged in vitro), the MuSCs rapidly inactivated for at least 7 days; a period 2× longer than previously possible (*Quarta et al., 2016*; *Charville et al., 2015*; *Arjona et al., 2022*). Analysis of MuSC activities reflected functional heterogeneity and population-level adaptations to achieve a steady-state equilibrium in pool size. Modulating various components of the niche revealed the unique setting of 3D extracellular matrix (ECM) with engineered myotubes to be vital to inactivation and sufficient for inducing in vivo-like hallmarks of quiescence never before reported in vitro, including cadherin-mediated niche interactions, elongated nuclei, and elaborated cytoplasmic projections (*Verma et al., 2018*; *Kann et al., 2022*; *Ma et al., 2022*; *Eliazer et al., 2019*). Integrating the culture assay with a high-content imaging system and CellProfiler-based image analysis pipelines allowed us to relate cell fate signatures to morphometric features and produced criteria to assess the quiescence of MuSCs based solely on morphology. Finally, aged MuSCs introduced into mini-IDLE displayed phenotypic and functional defects associated with their failure to properly inactivate, that were partially rescued by wortmannin, a treatment shown by others to push activated young MuSCs into a deeper quiescence. Thus, we present a new MuSC quiescence assay that recapitulates in vivo-like hallmarks of young and aged MuSCs within homeostatic muscle 'in a dish' for the first time, which enabled the identification of a putative strategy to correct aged MuSC dysfunction.

## Results

### Engineered myotube templates derived from primary mouse myoblasts maintain integrity for 2 weeks in culture

We first set out to engineer a skeletal muscle microenvironment suited to investigate the ability of a 3D myotube niche to induce a quiescent-like phenotype upon freshly isolated (i.e. activated) MuSCs cultured in vitro. We previously reported a method to prepare thin sheets of human myotubes situated within a 24-well format, together with a strategy to evaluate mouse MuSC endogenous repair 'in a dish' (*Davoudi et al., 2022*). Herein, we adapted and extended the method to create thin sheets of murine myotubes that fit within a 96-well plate footprint. Briefly, we incorporated primary mouse myoblasts within a mixture of media, fibrinogen, and Geltrex (i.e. reconstituted basement membrane proteins). The resultant slurry was pipetted into pieces of thin, porous cellulose teabag paper, pre-adsorbed with thrombin, and situated within a 96-well plate (*Figure 1A*). In this way, fibrin hydrogel gelation is delayed until the cell/fibrinogen slurry diffuses within the thrombin-containing cellulose scaffold. Following a 2-day equilibration period in growth media (GM), the tissues were transitioned to a low-mitogen differentiation media (DM) to support multinucleate myotube formation within the cellulose-reinforced fibrin hydrogel (*Figure 1A–C*).

Spontaneous twitch contractions were first observed on day 4 (data not shown). Peak myotube content (≈65% by sarcomeric α-actinin (SAA) tissue coverage) and a nuclear fusion index of 90% was achieved by 5 days in DM with as few as 25,000 cells per tissue (*Figure 1B–D*, *Figure 1—figure supplement 1*). Since myotube degradation could serve as an activation cue for the engrafted MuSCs, we evaluated the integrity of the tissues over time in culture. Starting on day 18, a visual inspection of tissues revealed loss of myotubes around the periphery of the tissues and quantification of SAA coverage showed a corresponding drop (*Figure 1B and D*). A colorimetric metabolic activity assay (i.e. MTS) revealed that mitochondrial activity was significantly reduced on day 18 when compared to day 10 (*Figure 1E*), another indication that the integrity of the tissues becomes compromised at later timepoints. Based on these analyses, we established conditions to engineer a mouse myotube template and concluded that day 5 to day 14 of myotube template culture would serve as the assay window.

### 3D myotube templates show improved maturity and integrity compared to 2D culture

Our prior studies comparing human myotubes cultivated in 2D and 3D settings revealed disadvantages of 2D culture that included myotube detachment from the substrate, which enriched for less mature myotubes over time (*Afshar Bakooshli et al., 2019*). We compared murine myotubes at days 5, 8, and 12 of differentiation in 2D and 3D culture, and drew similar conclusions. Specifically, by day 12 we observed detachment zones in 2D cultures which translated to a significant drop in SAA coverage compared to our 3D myotube templates (*Figure 1—figure supplement 2A and C*). Upon closer inspection, we also observed a higher incidence of multinucleated, SAA+ sphere-shaped structures, indicative of retracted myotubes (*Figure 1—figure supplement 2B and D*). To address myotube maturity, we evaluated SAA+ striations using a MATLAB z-disc analysis pipeline (*Morris et al., 2020*). According to the Z-line fraction (i.e. the fraction of SAA skeleton classified as Z-lines by actin guided segmentation), the proportion of striations per field of view is increased in 3D at the day 5 timepoint, with 2D cultures eventually catching up by day 8 (*Figure 1—figure supplement 2B and E*). While we noted no differences in mean continuous Z-line length, we found an increase in mean sarcomere length in 3D culture, which is closer to the length-tension plateau reported in vivo (*Figure 1—figure supplement 2F and G*; *Moo et al., 2016*). Thus, by contrast to 2D culture, 3D myotube templates show little to no myotube detachment over 12 days of differentiation and a quicker obtention of striations.

### MuSC populations persist in myotube templates

The engineered mouse myotube template incorporates key cellular, biochemical, and biophysical aspects of the MuSC niche: myofibers and ECM (*Mauro, 1961*; *Gilbert and Blau, 2011*; *Fuchs and Blau, 2020*). Thus, we next sought to determine whether adult mouse MuSCs could persist, in terms of pool size and Pax7 expression, when introduced to these biomimetic cultures. Firstly, we adapted a

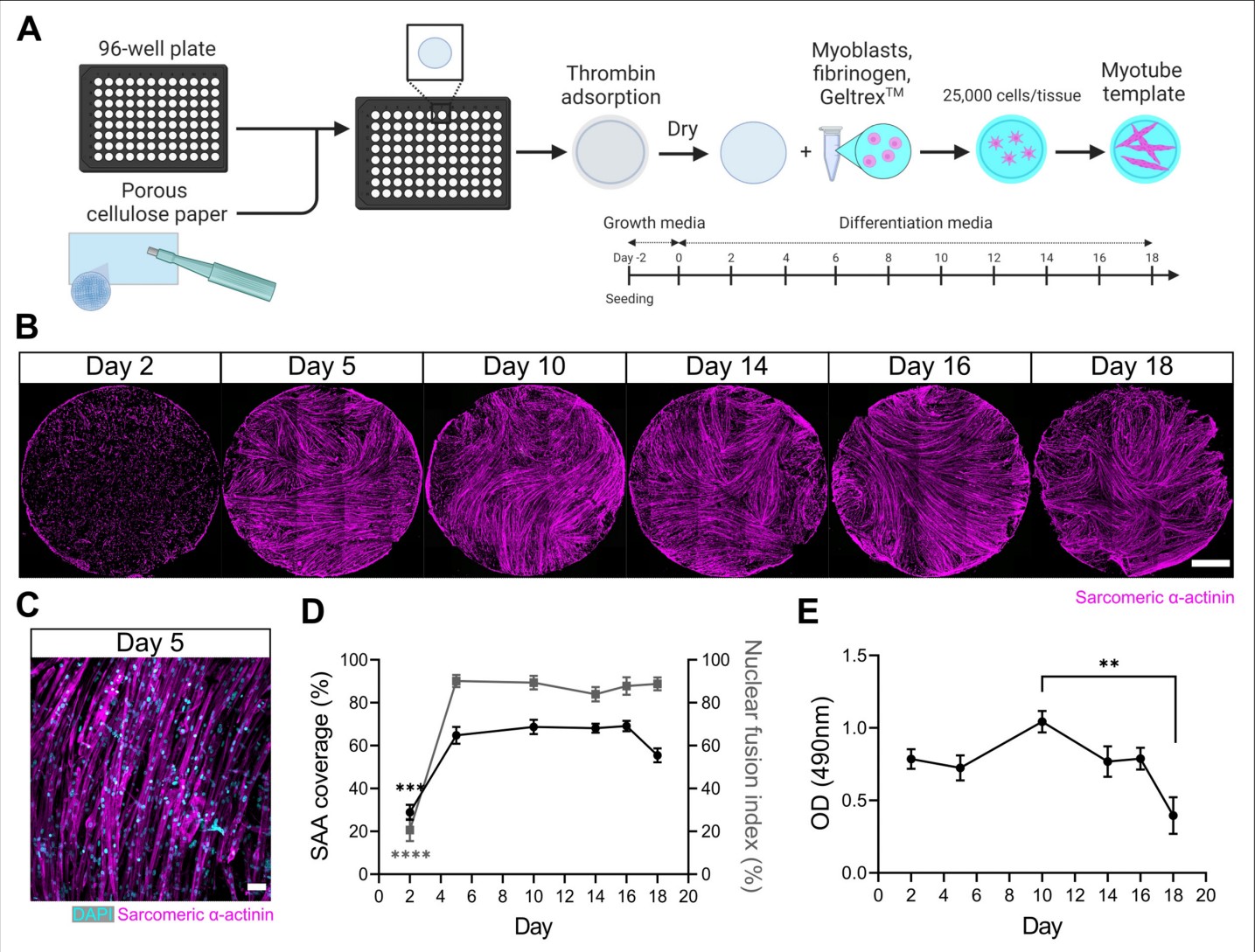

**Figure 1.** A three-dimensional (3D) murine skeletal muscle myotube template with a 96-well footprint. (**A**) Schematic overview of the strategy used to generate myotube templates with an associated timeline for downstream culture (made with BioRender). (**B**) Representative confocal stitched images of myotube templates labelled for sarcomeric α-actinin (SAA; magenta) at days 2, 5, 10, 14, 16, and 18 of culture. Scale bar, 1 mm. (**C**) Representative confocal image of myotubes at day 5 labelled with DAPI (cyan) and SAA (magenta). Scale bar, 50 μm. (**D**) Quantification of SAA area coverage (left axis; black line) and nuclear fusion index (right axis; grey line) of myotube templates at days 2, 5, 10, 14, 16, and 18 of culture. n=9–16 across N=3–6 independent biological replicates. Graph displays mean ± s.e.m.; one-way ANOVA with Tukey's post-test, minimum ***p=0.002 (SAA coverage), ****p<0.0001 (nuclear fusion index). (**E**) Optical density (OD) at 490 nm of media after myotube template incubation with MTS assay reagent on days 2, 5, 10, 14, 16, and 18 of culture. n=9–12 across N=3–4 independent biological replicates. Graph displays mean ± s.e.m.; one-way ANOVA with Tukey's post-test, **p=0.0033. Raw data available in *Figure 1—source data 1*.

The online version of this article includes the following source data and figure supplement(s) for figure 1:

**Source data 1.** Raw data for *Figure 1*.

**Figure supplement 1.** Optimization of cell seeding density.

**Figure supplement 1—source data 1.** Raw data for *Figure 1—figure supplement 1*.

**Figure supplement 2.** Accelerated maturity and prolonged culture of myotube templates over two-dimensional (2D) monolayers.

**Figure supplement 2—source data 1.** Raw data for *Figure 1—figure supplement 2*.

magnetic-activated cell sorting (MACS) protocol as a convenient and fast alternative to fluorescence-activated cell sorting (FACS) for enriching the Pax7+ mononucleated cell population from digested skeletal muscle. By conducting two rounds of microbead-based lineage depletion followed by integrin α-7 enrichment, we achieved an average purity of 93% Pax7+ cells (*Figure 2—figure supplement 1*),

which meets FACS purity values reported by others (*Maesner et al., 2016*; *Kuang et al., 2007*). Using this protocol, Pax7$^+$ MuSCs were enriched from the enzymatically dissociated hindlimb muscles of 129-Tg(CAG-EYFP)7AC5Nagy/J transgenic mice (*Hadjantonakis et al., 2002*). Freshly sorted MuSCs were seeded onto day 5 myotube templates, and the tissue co-cultures processed for analysis at 1, 3, and 7 days post-engraftment (DPE) (*Figure 2A*). Over the 1-week culture period, the Pax7$^+$ mononuclear donor (YFP$^+$) cells were seen distributed throughout the myotube template and adopting an elongated morphology that aligned with the local myotubes (*Figure 2B–C*). We investigated the effect of introducing different numbers of MuSCs onto individual myotube templates, by quantifying the population of Pax7$^+$ mononuclear donor cells over time. Seeding 500 MuSCs resulted in a relatively stable pool size over time. Interestingly, when a higher (1500 or 2500) or lower number of MuSCs were introduced to myotube templates, over time the number of Pax7$^+$ donor cells converged to match the pool size attained in the 500 MuSC condition (*Figure 2D*). Collectively, these data indicate that the engrafted MuSC population persists and establishes a steady-state cell density within the engineered niche.

## MuSCs reversibly inactivate within myotubes templates

We next studied the behaviour and fate of freshly isolated MuSCs engrafted within myotube templates and determined that they inactivate over a 7-day culture period, and can be coaxed to reactivate by injury stimuli. We began by evaluating MuSCs within the engraftment condition that lent to a stable population density over time (i.e. 500 MuSCs per tissue). Calcitonin receptor (CalcR) expression is a hallmark of quiescent MuSCs (*Yamaguchi et al., 2015*; *Baghdadi et al., 2018*; *Robinson et al., 2021*; *Gnocchi et al., 2009*). Indeed, at the protein level, CalcR is expressed by quiescent MuSCs, but is then absent from all MuSCs within 48 hr of an in vivo myotoxin injury or within 48 hr of prospective isolation followed by in vitro culture (*Yamaguchi et al., 2015*; *Baghdadi et al., 2018*; *Gnocchi et al., 2009*). In the context of our 3D culture assay, the majority of MuSCs expressed CalcR at 1 DPE, with a sharp decline in the proportion of CalcR$^+$ donor cells observed at 3 DPE (*Figure 3—figure supplement 1*). Interestingly, ~15% of donor MuSCs were CalcR$^+$ at both 3 and 7 DPE (*Figure 3—figure supplement 1*). Given the lack of evidence for CalcR$^+$ MuSCs in prolonged in vitro cultures, we posited that this subpopulation might be reflective of MuSCs that had resisted activation in favour of maintaining a more quiescent-like state, which we sought to interrogate further.

After a single day of culture, we found that ≈75% of the donor MuSCs (YFP$^+$caveolin-1$^+$ cells) engrafted within the myotube templates expressed the transcription factor c-FOS, among the earliest transcriptional events reported to date in the MuSC activation sequence (*Machado et al., 2017*; *Almada et al., 2021*; *Yue et al., 2020*; *van Velthoven et al., 2017*). The existence of c-FOS$^-$ donor cells at this timepoint is consistent with the notion of an activation refractory subpopulation. By 3 DPE, the proportion of caveolin-1$^+$c-FOS$^+$ mononuclear cells dropped to ≈30%, with similar proportions observed on 7 DPE (*Figure 3A–B*). The maintenance of a steady-state population of donor MuSCs from 1 to 3 DPE, coupled with the rapid loss of c-FOS immunolabelling by 3 DPE, suggests that myotube template culture induces MuSCs to inactivate.

Consistently, when we quantified the incidence of MuSCs in the active phase of the cell cycle via Ki67 labelling, we found that at 3 DPE, only ≈1/3 of the Pax7$^+$ mononuclear donor cell population was Ki67$^+$, and this dropped to ≈10% by 7 DPE (*Figure 3C*). To better resolve the proliferative trajectory of the engrafting MuSCs, we conducted a Ki67 co-labelling study whereby 5-ethanyl-2'-deoxyuridine (EdU) was refreshed in the culture media daily over the 1-week culture period (*Figure 3D*). Of the Ki67$^-$ mononuclear donor cells present at 7 DPE, the vast majority were EdU$^-$ (*Figure 3E*). Approximately 30% were EdU$^+$, indicating cell-cycle entry at some point during the 1-week culture period, and a cessation by 7 DPE (*Figure 3E*). This correlates well with the proportion of Ki67$^+$ MuSCs we observed at 3 DPE (*Figure 3B*). This data, together with a scarcity of EdU$^+$ myonuclei observed in the cultures (data not shown), suggests that the main fate of the EdU labelled MuSCs is eventual cell-cycle exit, and not myotube fusion.

Lastly, we sought to understand whether the inactivated donor cells at 7 DPE were capable of re-entering the cell cycle. We first established a barium chloride exposure protocol that induced effective clearing of the myotubes with a non-significant change to MuSC population density (*Figure 3F* and *Figure 3—figure supplement 2*). We then analysed the mononuclear YFP$^+$Pax7$^+$ population 2 days post-injury and observed a statistically significant increase in the proportion of Ki67$^+$ cells as

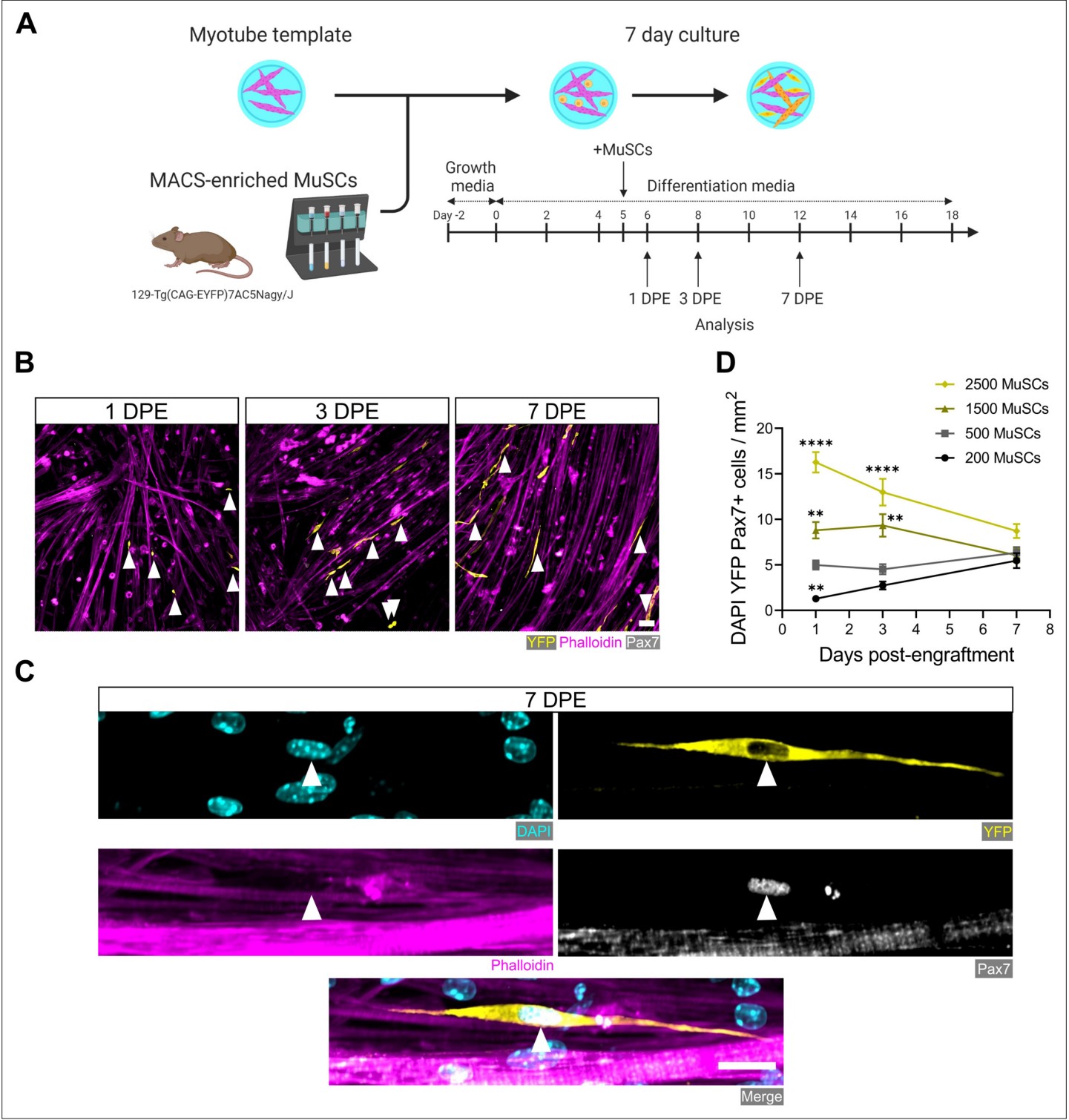

**Figure 2.** Engrafted muscle stem cells (MuSCs) persist in myotube template cultures and achieve a steady-state population density. (**A**) Schematic overview of the engraftment of freshly isolated MuSCs and the timeline for downstream analysis (made with BioRender). (**B**) Representative confocal images of myotube templates (phalloidin: magenta) with engrafted MuSCs (YFP: yellow, Pax7: white, white arrows) at 1, 3, and 7 days post-engraftment (DPE). Scale bar, 50 μm. (**C**) Representative confocal image of a donor MuSC (DAPI: cyan, YFP: yellow, Pax7: white) indicated with a white arrow, and myotubes (phalloidin: magenta) at 7 DPE. Scale bar, 20 μm. (**D**) Quantification of mononuclear DAPI⁺YFP⁺Pax7⁺ cell density per mm² at 1, 3, and 7 DPE across different starting MuSC engraftment numbers (200, 500, 1500, and 2500). n=9–15 across N=3–5 independent biological replicates. Graph displays

*Figure 2 continued on next page*

*Figure 2 continued*

mean ± s.e.m.; one-way ANOVA with Dunnet's test for each individual timepoint comparing against the 500 MuSC condition, **p=0.0025, 0.0051, 0.0029, ****p<0.0001. Raw data available in *Figure 2—source data 1*.

The online version of this article includes the following source data and figure supplement(s) for figure 2:

**Source data 1.** Raw data for *Figure 2*.

**Figure supplement 1.** Population purity in magnetic-activated cell sorting (MACS) isolated muscle stem cells (MuSCs).

**Figure supplement 1—source data 1.** Raw data for *Figure 2—figure supplement 1*.

compared to the control condition (*Figure 3G*). Thus, myotube template cultures allow for inactivation and cell-cycle exit of engrafted MuSCs, which can be reversed with the injury-associated stimuli caused by barium chloride exposure.

## Engrafted MuSCs adapt their pool size to a myotube template threshold

Regardless of the initial size of the MuSC pool, a common mononuclear YFP⁺Pax7⁺ cell density was attained by 7 DPE (*Figure 2D*). To uncover cellular mechanisms underlying the acquisition of a MuSC steady-state population density, we investigated how the donor MuSC pool responded under a set of distinct starting conditions. We began by extending the EdU/Ki67 co-labelling study (*Figure 3D–E*) to include an evaluation of conditions where more (1500, 2500) or less (200) MuSCs were seeded onto the myotube templates. Compared with the 500 MuSC seeding condition, we found a significant increase in the proportion of mononuclear YFP⁺Ki67⁻ cells that were EdU⁺ at 7 DPE in cultures seeded with 200 MuSCs, suggesting the MuSC pool expanded to attain a steady-state density (*Figure 3—figure supplement 3A and C*). By contrast, in conditions where >500 MuSCs were seeded, a significant decrease in the proportion of mononuclear YFP⁺Ki67⁻ cells that were EdU⁺ at 7 DPE was observed (*Figure 3—figure supplement 3C*). In these conditions, a decrease in the MuSC pool size by 7 DPE could be achieved through cell death or by fusion into myotubes. Consistent with the latter hypothesis, upon visual inspection we saw a qualitatively greater number of donor-derived myotubes in the cultures seeded with >500 MuSCs (*Figure 3—figure supplement 3B*), which was confirmed by quantifying the percentage area of myotube templates covered by YFP signal (*Figure 3—figure supplement 3D*). In sum, we conclude that MuSCs meet a steady-state population density via increased proliferation when beginning below the 500 cell threshold, and with increased cell fusion when beginning above it.

## A 3D myotube culture is required for a persistent MuSC population in vitro

The rapid inactivation and subsequent maintenance of Pax7⁺ MuSCs engrafted within the 3D myotube templates (*Figures 2–3*) represents a divergent phenotype when compared to conventional 2D culture (*Figure 4A–B*; *Gilbert et al., 2010*). Therefore, we next sought to elucidate culture design criteria that served to support MuSC inactivation and pool maintenance over time. We first explored the response of MuSCs seeded onto tissues on day 0 of myotube template differentiation, a timepoint corresponding to the earliest myocyte fusion events, and therefore when myotubes were absent from the tissues. Compared to MuSCs seeded on myotube templates on day 5 of differentiation, day 0 seeding resulted in a progressive loss of YFP⁺Pax7⁺ mononuclear cells, and most of those that remained were Ki67⁺ (*Figure 4C–D*). The striking contrast in YFP⁺ myotube content observed at 7 DPE upon comparing these two conditions suggests that the MuSCs engrafted on day 0 had undergone differentiation (*Figure 4—figure supplement 1*). We next determined whether a 3D cellulose-reinforced hydrogel alone was sufficient to support MuSC inactivation and maintenance, since the myocytes present on day 0 of differentiation may have exerted a dominant effect overriding contributions of the 3D culture environment. However, this notion was abandoned upon finding that the outcome of this culture scenario (*Figure 4E*) very closely matched what we observed when the MuSCs were cultured in 2D Geltrex-coated culture wells (*Figure 4B*); a loss of the YFP⁺Pax7⁺ mononuclear population over time. Our results instead seemed to suggest that the myotube template played a central role in inactivating and maintaining a persistent population of MuSCs in culture. Indeed, upon adding MuSCs to a day 5 monolayer of myotubes in 2D culture, a Pax7⁺ population was maintained

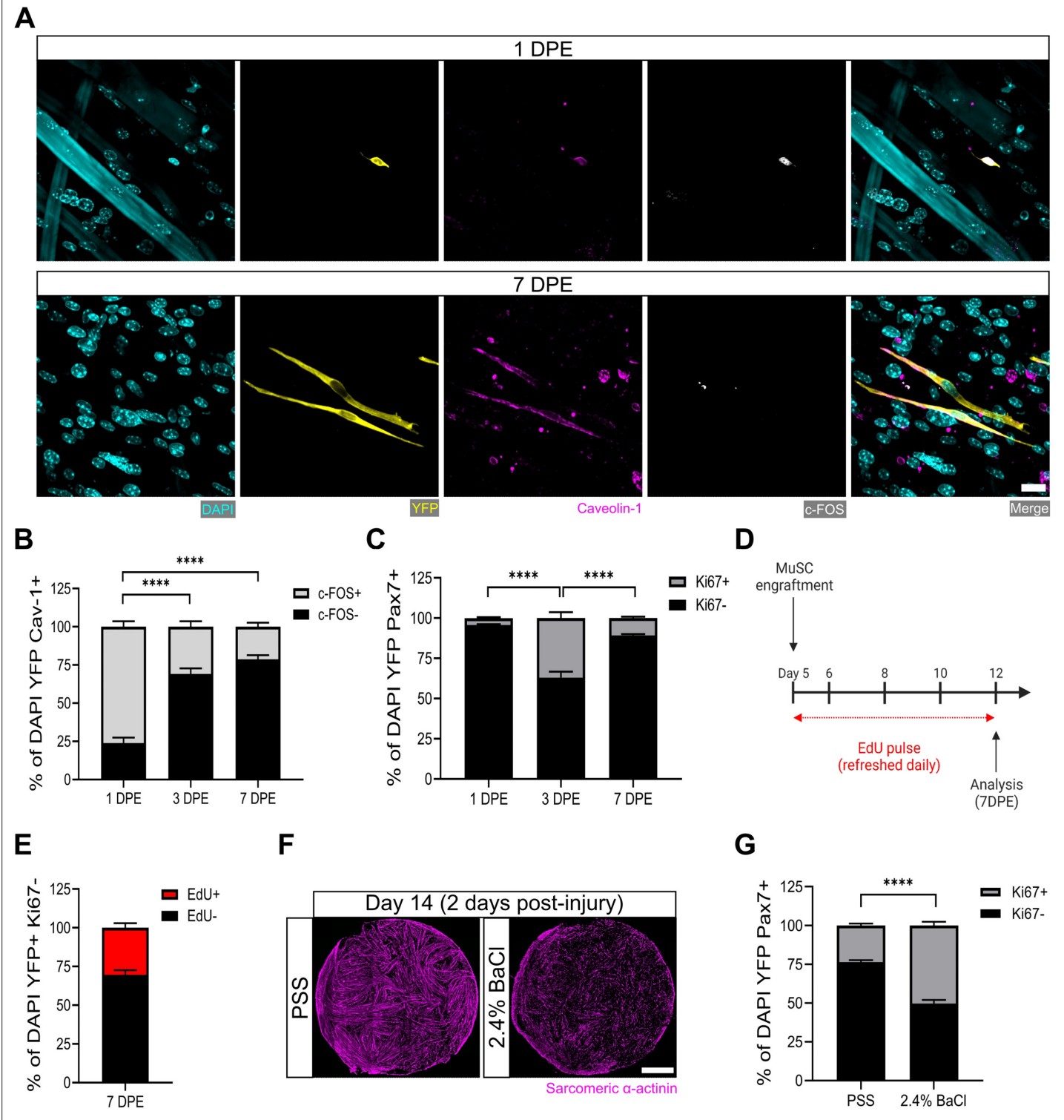

**Figure 3.** Muscle stem cells (MuSCs) engrafted within engineered muscle tissue exit cell cycle and inactivate. (**A**) Representative confocal image of a mononuclear cell (DAPI: cyan) positive for YFP (yellow), caveolin-1 (magenta), and c-FOS (white) at 1 day post-engraftment (DPE) (top), and a c-FOS⁻ cell at 7 DPE (bottom). Scale bar, 20 μm. (**B**) Stacked bar graph showing proportions of c-FOS ± cells at 1, 3, and 7 DPE in the DAPI⁺YFP⁺Cav-1⁺ population. n=9 across N=3 independent biological replicates. Graph displays mean ± s.e.m. for c-FOS⁺ and c-FOS⁻; one-way ANOVA with Tukey's post-test comparing the FOS⁻ proportions of each timepoint, ****p<0.0001. (**C**) Stacked bar graph showing proportions of Ki67 ± cells at 1, 3, and 7 DPE in the DAPI⁺YFP⁺Pax7⁺ population. n=10–11 across N=3–4 independent biological replicates. Graph displays mean ± s.e.m. for Ki67⁺ and Ki67⁻; one-way ANOVA with Tukey's post-test comparing the Ki67⁻ proportions of each timepoint, ****p<0.000.1. (**D**) Timeline of EdU/Ki67 co-labelling experiment

*Figure 3 continued on next page*

*Figure 3 continued*

(made with BioRender). (**E**) Stacked bar graph showing proportions of EdU ± cells at 7 DPE in the DAPI$^+$YFP$^+$Ki67$^-$ mononuclear cell population. n=15 across N=5 independent biological replicates. Graph displays mean ± s.e.m. for EdU$^+$ and EdU$^-$. (**F**) Representative confocal stitched images of myotube templates (sarcomeric α-actinin (SAA): magenta) 2 days after a 4 hr exposure to the physiological salt solution (PSS) control or a 2.4% barium chloride (BaCl$_2$) solution. Scale bar, 1 mm. (**G**) Proportion of Ki67 ± cells at 2 days post-injury (DPI) in the DAP$^+$YFP$^+$Pax7$^+$ population. n=16, 18 across N=5, 6 biological replicates. Graph displays mean ± s.e.m. for Ki67$^+$ and Ki67$^-$; unpaired t-test of the Ki67$^-$ proportions of both conditions, ****p<0.0001. Raw data available in *Figure 3—source data 1*.

The online version of this article includes the following source data and figure supplement(s) for figure 3:

**Source data 1.** Raw data for *Figure 3*.

**Figure supplement 1.** Persistent CalcR$^+$ population amongst Pax7$^+$ donor cell population at 7 days post-engraftment (DPE).

**Figure supplement 1—source data 1.** Raw data for *Figure 3—figure supplement 1*.

**Figure supplement 2.** Characterization of barium chloride-induced injury.

**Figure supplement 2—source data 1.** Raw data for *Figure 3—figure supplement 2*.

**Figure supplement 3.** Regulation of muscle stem cell (MuSC) pool size in myotube template cultures.

**Figure supplement 3—source data 1.** Raw data for *Figure 3—figure supplement 3*.

over the 1-week culture period (*Figure 4F*). But in striking contrast to the 3D myotube template culture (*Figure 4C*), only a minority of the Pax7$^+$ donor cells were Ki67$^-$ at 7 DPE (*Figure 4F–G*). From this iterative analysis, we conclude that myotubes are necessary for Pax7$^+$ MuSC persistence, and that the combination of myotubes and a 3D culture environment drives the MuSC inactivation process.

## Engrafted MuSCs adopt quiescent-like morphologies that predict cell fate signature

Qualitatively, the engrafted MuSCs in our cultures adopted an elongated morphology over time (*Figure 2B–C*, *Figure 3A*, *Figure 5A*), reminiscent of quiescent MuSCs in vivo, and contrasting against morphologies observed in 2D cultures (*Figure 5—figure supplement 1A*; *Verma et al., 2018*; *Kann et al., 2022*; *Ma et al., 2022*). To quantify MuSC morphogenic progression in culture, we next overcame a significant data analysis bottleneck by establishing and validating a CellProfiler-based image analysis pipeline in order to segment and evaluate donor MuSCs in our phenotypic datasets (see Materials and methods and *Figure 5—figure supplement 2*; *Stirling et al., 2021*). The cytoplasmic elongation of mononucleated Pax7$^+$ donor cell bodies was captured by applying a ratio of max/min Feret diameter to segmented images of tissues immunostained for YFP, Pax7, and DAPI. The roundness of nuclei within mononucleated YFP$^+$Pax7$^+$ cells was evaluated using the measurement of eccentricity, whereby a value of 0 corresponds to a perfect circle, and a value of 1 to a straight line (*Figure 5B*). With this pipeline, we determined that the Pax7$^+$ donor cell population progressively shifted from low max/min Feret diameter ratios and eccentricities (lower left quadrant) to high max/min Feret diameter ratios and eccentricities (upper right quadrant) over time in 3D myotube culture (*Figure 5C*). By conducting a side-by-side comparison of these features when MuSCs were cultured in 2D culture, vs. amongst a 2D myotube monolayer or within the 3D myotube template, we concluded that quiescence-associated morphologies were most prominent in the context of the 3D engineered niche (*Figure 5—figure supplement 1B–C*). Subsequently, we correlated MuSC morphology against another feature of quiescent MuSCs, the accumulation of DDX6$^+$ cytoplasmic mRNP granules, which were previously shown to sequester activation-specific transcripts (*Crist et al., 2012*). Consistently, DDX6$^+$ puncta progressively accumulate within MuSCs cultured in the 3D in vitro assay (*Figure 5D–E*), and we report a significant linear trend increase between the morphology of the cells and the number of DDX6$^+$ puncta at 7 DPE (*Figure 5F*). Thus, the rice-like nuclear morphology and elaborated cytoplasmic projections of the Pax7$^+$ donor cells at 7 DPE resembled quiescent features of MuSCs in vivo (*Verma et al., 2018*; *Kann et al., 2022*; *Ma et al., 2022*) and associate with a known molecular feature of quiescent cells.

We next sought to determine whether the donor MuSC morphologies observed in our cultures corroborated with the activation status of the cells. We introduced immunolabelling for MyoD, which, together with Pax7 staining, delivered molecular signatures for activated (Pax7$^+$MyoD$^+$) and inactivated (Pax7$^+$MyoD$^-$) donor cell populations. As expected, the ratio of Pax7$^+$MyoD$^+$ to Pax7$^+$MyoD$^-$ donor cells over time followed a trend similar to Ki67 status (*Figure 3C*), with a transient increase

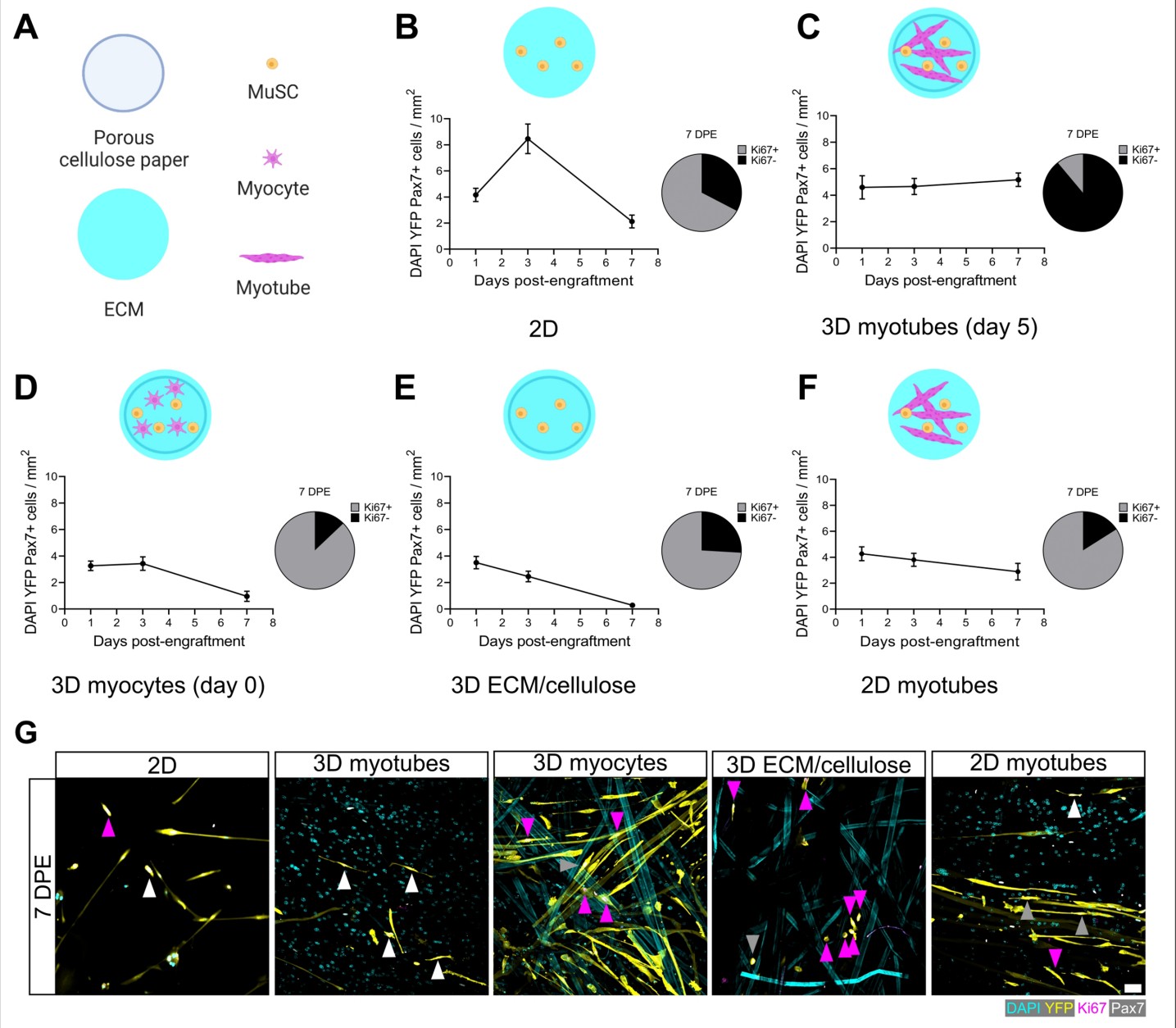

**Figure 4.** Permissive culture conditions for a persistent muscle stem cell (MuSC) population in vitro. (**A**) Key for figure icons. (**B–F**) Line graphs of mononucleated DAPI$^+$YFP$^+$Pax7$^+$ cell density at 1, 3, and 7 days post-engraftment (DPE) (left) and pie charts showing the proportion of Ki67 ± cells at 7 DPE (right) for cells seeded into a two-dimensional (2D) microwell with a Geltrex coating (**B**), engrafted into 3D myotube templates on day 5 (**C**) vs. day 0 (**D**) of differentiation. Additional comparisons include engraftment into a 3D cellulose-reinforced extracellular matrix (ECM) hydrogel on day 5 (**E**), or onto a 2D monolayer of myotubes with a Geltrex undercoating on day 5 of differentiation (**F**). n=6–15 from N=2–3 independent biological replicates. Graphs display mean ± s.e.m. (**G**) Representative confocal images of YFP$^+$ (yellow) donor cells (DAPI: cyan) at 7 DPE engrafted in 2D, 3D with myotubes (day 5), 3D with myocytes (day 0), 3D with cellulose-reinforced ECM or with a 2D monolayer of myotubes. Cells are also labelled for Ki67 (magenta) and Pax7 (white) where Ki67$^-$Pax7$^+$ cells are indicated with white arrows, Ki67$^+$Pax7$^+$ with grey arrows, and Ki67$^+$Pax7$^-$ with magenta arrows. Scale bar, 50 μm. Raw data available in *Figure 4—source data 1*.

The online version of this article includes the following source data and figure supplement(s) for figure 4:

**Source data 1.** Raw data for *Figure 4*.

**Figure supplement 1.** Increased YFP coverage when muscle stem cells (MuSCs) engrafted on day 0 of myotube template differentiation.

**Figure supplement 1—source data 1.** Raw data for *Figure 4—figure supplement 1*.

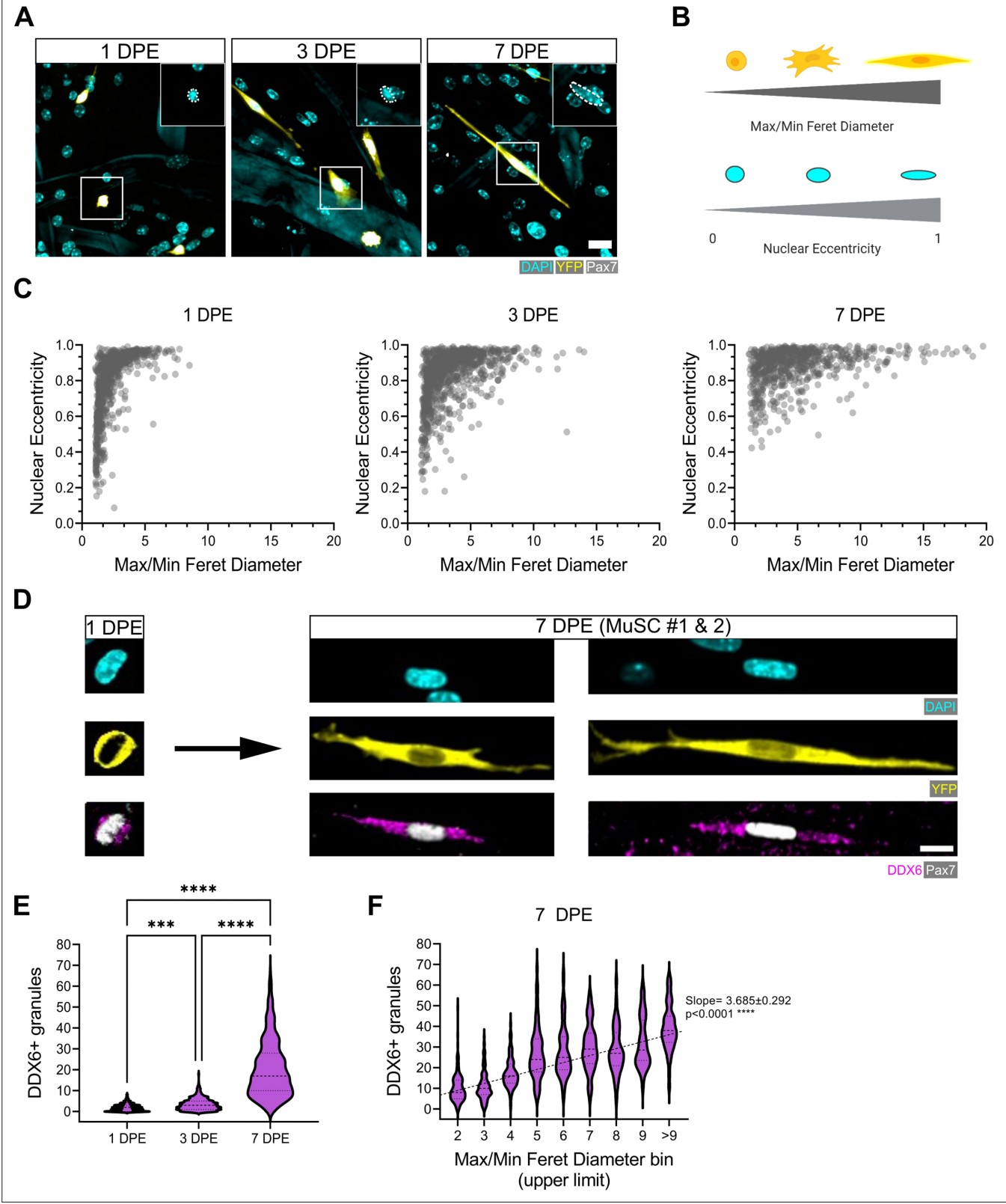

**Figure 5.** Morphological evolution of engrafted muscle stem cells (MuSCs). (**A**) Representative confocal images of MuSCs (DAPI: cyan, YFP: yellow, Pax7: white) with distinct morphological features at 1, 3, and 7 days post-engraftment (DPE). Insets highlight nuclear morphology with a white dotted outline. Scale bar, 20 μm. (**B**) Schematic demonstrating the morphological features quantified using CellProfiler (made with BioRender). (**C**) Dot plot graphs showing individual Pax7⁺ donor cells and their associated max/min Feret diameter ratio and nuclear eccentricity at 1 (left), 3 (middle), and 7 DPE

*Figure 5 continued on next page*

*Figure 5 continued*

(right). n=916, 980, and 737 across N=3–4 biological replicates. (**D**) Representative confocal images of MuSCs (DAPI: cyan, YFP: yellow, Pax7: white) labelled for p54/RCK (DDX6) at 1 and 7 DPE. Scale bar, 10 µm. (**E**) DDX6$^+$ granule quantification in individual MuSCs at 1, 3, and 7 DPE. n=639, 770, and 676 across N=3 independent biological replicates. Graph displays violin plot distribution; one-way ANOVA with Tukey's post-test, ***p=0.0004, ****p<0.0001. (**F**) Violin plot distribution of DDX6$^+$ granules in individual MuSCs at 7 DPE stratified across max/min Feret diameter bins. n=737 across N=3 independent biological replicates. One-way ANOVA with test for linear trend across bins, ****p<0.0001. Raw data available in *Figure 5—source data 1*.

The online version of this article includes the following source data and figure supplement(s) for figure 5:

**Source data 1.** Raw data for *Figure 5*.

**Figure supplement 1.** Pax7$^+$ donor cell morphologies in two (2D) and three dimensions (3D).

**Figure supplement 1—source data 1.** Raw data for *Figure 5—figure supplement 1*.

**Figure supplement 2.** CellProfiler pipeline for muscle stem cell (MuSC) identification and characterization.

**Figure supplement 2—source data 1.** Raw data for *Figure 5—figure supplement 2*.

**Figure supplement 3.** Morphological characterization of Pax7$^+$MyoD and Pax7$^+$MyoD$^+$ muscle stem cells (MuSCs).

**Figure supplement 3—source data 1.** Raw data for *Figure 5—figure supplement 3*.

**Figure supplement 4.** ROCK inhibition hastens muscle stem cell (MuSC) inactivation.

**Figure supplement 4—source data 1.** Raw data for *Figure 5—figure supplement 4*.

**Figure supplement 5.** ROCK inhibition confers acquisition of quiescent-like morphologies to the Pax7+MyoD+ population.

**Figure supplement 5—source data 1.** Raw data for *Figure 5—figure supplement 5*.

in Pax7$^+$MyoD$^+$ cells at 3 DPE and a predominance of Pax7$^+$MyoD$^-$ cells at 7 DPE (*Figure 5—figure supplement 3A–B*). By evaluating the mean max/min Feret diameter ratio and mean eccentricity values of the Pax7$^+$MyoD$^+$ and Pax7$^+$MyoD$^-$ cell populations, we found that nuclear eccentricity differs between the populations by 3 DPE, while population divergence according to max/min Feret diameter ratio (>2-fold) emerged a bit later, at 7 DPE (*Figure 5—figure supplement 3C*). Specifically, nuclear morphology of the Pax7$^+$MyoD$^-$ population showed a progressive, statistically significant transition to a rice-like shape, while Pax7$^+$MyoD$^+$ nuclei remained more rounded. Elongation of the cell body and elaborate projections were features that exclusively characterized the Pax7$^+$MyoD$^-$ cell population, and emerged between 3 and 7 DPE. Indeed, donor cells with a max/min Feret diameter ≥5.8 uniformly displayed the Pax7$^+$MyoD$^-$ signature of inactivated cells (*Figure 5—figure supplement 3A*).

Quiescence-associated MuSC morphologies were recently shown to be induced and maintained by tipping the Rho family GTPase balance to favour cytoskeletal remodelling events caused by Rac signalling (*Kann et al., 2022*). Hence, we asked whether inhibiting Rho signalling could expedite the process of MuSC inactivation and the emergence of quiescence-associated features. We added the Y-27632 ROCK inhibitor to the culture media for 1 week after MuSC engraftment. The treatment did not result in a significant shift in Pax7$^+$ cell population-level kinetics (*Figure 5—figure supplement 4A–B*), and while c-FOS labelling showed a trending decrease in c-FOS$^+$ cells at 3 DPE, it was not significant (*Figure 5—figure supplement 4C*). However, MyoD labelling revealed a Y-27632-induced block on the acquisition of a transiently activated subpopulation of MuSCs as had been observed in the control conditions at 3 DPE (*Figure 5—figure supplement 4D*). This was coupled to clear changes to MuSC morphology emerging at the 3 DPE timepoint (*Figure 5—figure supplement 4A*). Specifically, our evaluations of the max/min Feret diameter ratios of MuSCs indicated that Y-27632 treatment caused a shift towards the elongated morphology of quiescent-like cells at 3 DPE that was not attained in the control condition until 7 DPE. Thus, the Y-27632 treatment appeared to hasten the process of inactivation and, also, the acquisition of a quiescent-like morphology. Next, we quantified max/min Feret diameter ratio, nuclear eccentricity, mean number of cytoplasmic branches per cell, and mean skeleton length per cell and related these metrics to the Pax7$^+$MyoD$^-$ and Pax7$^+$MyoD$^+$ signature of each donor cell at 1, 3, and 7 DPE (*Figure 5—figure supplement 5*). From this, we observed that the Y-27632 treatment elicited a significant influence over the morphology of the Pax7$^+$MyoD$^+$ population, which in some cases (e.g. nuclear eccentricity, branching), matched, but did not supersede, the quiescent-like morphology of the Pax7$^+$MyoD$^-$ engrafted cells. By contrast, the progression of the Pax7$^+$MyoD$^-$ population towards a quiescent-like morphology appeared to be uninfluenced by the Y-27632 treatment when compared against the control. Thus, Y-27632 treatment increased the

proportion of inactivated (c-Fos⁻Pax7⁺MyoD⁻) MuSCs at 3DPE, which progressed to a quiescent-like morphology in a Y-27632 agnostic manner. In parallel, Y-27632 treatment coaxed the Pax7⁺MyoD⁺ population to acquire quiescence-associated morphologies.

Taken together, morphological analysis of the engrafted MuSCs suggests that changes in nuclear morphology precede cell body extension and establishment of quiescent-like projections during the inactivation process, that morphometric features alone may predict MuSC inactivation status, and offers further support that cytoskeletal remodelling through Rho family GTPase signalling dictates the MuSC transition between activation and quiescence.

## Engrafted MuSCs establish a polarized niche

We next evaluated additional hallmarks of quiescent MuSCs including the spatial organization of cadherins, integrins, and ECM proteins relative to their niche (*Kuang et al., 2008*; *Bröhl et al., 2012*; *Goel et al., 2017*). In vivo, MuSCs are identified anatomically by their positioning sandwiched between a myofiber and the surrounding basal lamina (*Mauro, 1961*). This polarized niche lends to the intracellular segregation or deposition of proteins within MuSCs to the apical side facing the myofiber (e.g. M-cadherin) or to the basal side facing the basal lamina (e.g. integrin α-7, laminin) (*Bröhl et al., 2012*; *Goel et al., 2017*). By evaluating immunolabelled tissues at 7 DPE, we found that most mononucleated donor cells had an elongated morphology and were closely associated with multinucleated myotubes. For more than two-thirds of donor cells, M-cadherin expression restricted to the apical interface was observed (*Figure 6—figure supplement 1*).

It was recently discovered that quiescent MuSCs localize the N-cadherin adhesion molecule to the tips of elaborated cytoplasmic projections (coined 'quiescent projections') (*Kann et al., 2022*), a feature we also observed within our culture assay at 7 DPE (*Figure 6A*). Examples such as these were found in ≈45% of mononuclear donor cells (*Figure 6C*), and in each case the Pax7⁺ donor cell morphology was characterized by a long oval-shaped nucleus and very long, elaborated cytoplasmic projections. Furthermore, we identified examples of polarized distribution of M-cadherin and integrin α-7 or laminin α-2 in Pax7⁺ donor cells, albeit with lower frequency (*Figure 6B and D* and *Figure 6—figure supplement 2*). This evidence demonstrates that the engrafted MuSCs can recapitulate anatomical hallmarks of MuSCs residing within adult homeostatic skeletal muscle, and suggests that acquisition of these features is dependent on interactions with MuSCs and their immediate myofiber niche.

## Aged MuSCs exhibit delayed inactivation in mini-IDLE that is rescuable by Akt inhibition

We have shown that freshly isolated MuSCs are coaxed into a quiescent-like state in mini-IDLE that is characterized by cell-cycle exit, a Pax7⁺DDX6⁺MyoD⁻c-Fos⁻ signature, morphological and niche-associated features. A hallmark of MuSCs residing within aged muscle is precocious activation, owing to an improper maintenance and/or return to a quiescent state (*Chakkalakal et al., 2012*; *Haroon et al., 2022*; *García-Prat et al., 2020*; *Kimmel et al., 2020*; *Kimmel et al., 2021*). Indeed, a proportion of aged MuSCs remain in a state of chronic activation (*Dong et al., 2022*). As a result of the improper repair kinetics caused by the 'activated' state, these MuSCs fail to meet regenerative demand and are eventually depleted with further age (*Chakkalakal et al., 2012*; *Cosgrove et al., 2014*). We next leveraged mini-IDLE to evaluate possible defects in aged MuSC quiescence that might be apparent when they are decoupled from an aged niche environment. Upon seeding 500 MuSCs isolated from aged muscle onto a young 3D myotube template, we quantified an ≈2-fold increase in Pax7⁺ mononucleated donor cell density by 3 DPE relative to tissues engrafted by young MuSCs and analysed at the same timepoint (*Figure 7A* and *Figure 7—figure supplement 1A*). Consistently, a greater proportion of aged as compared to young MuSCs were Ki67⁺ at 3 DPE, and aged MuSC morphology at this timepoint diverged significantly from that observed of young MuSCs in 3D myotube cultures (*Figure 7B–C* and *Figure 2B*). By 7 DPE, aged MuSC engrafted cultures showed a small, but significant, decrease in population density compared to young MuSC engrafted tissues (*Figure 7A*). This, coupled with the trending increase in donor cell GFP⁺ signal covering tissues at this timepoint (*Figure 7—figure supplement 2*), suggested that the aged MuSCs were unable to maintain pool size and that the production of Pax7⁺ donor cells observed at 3 DPE culminated in differentiation to myotubes. Nonetheless, the mononucleated aged Pax7⁺ donor cell population that persisted throughout the culture

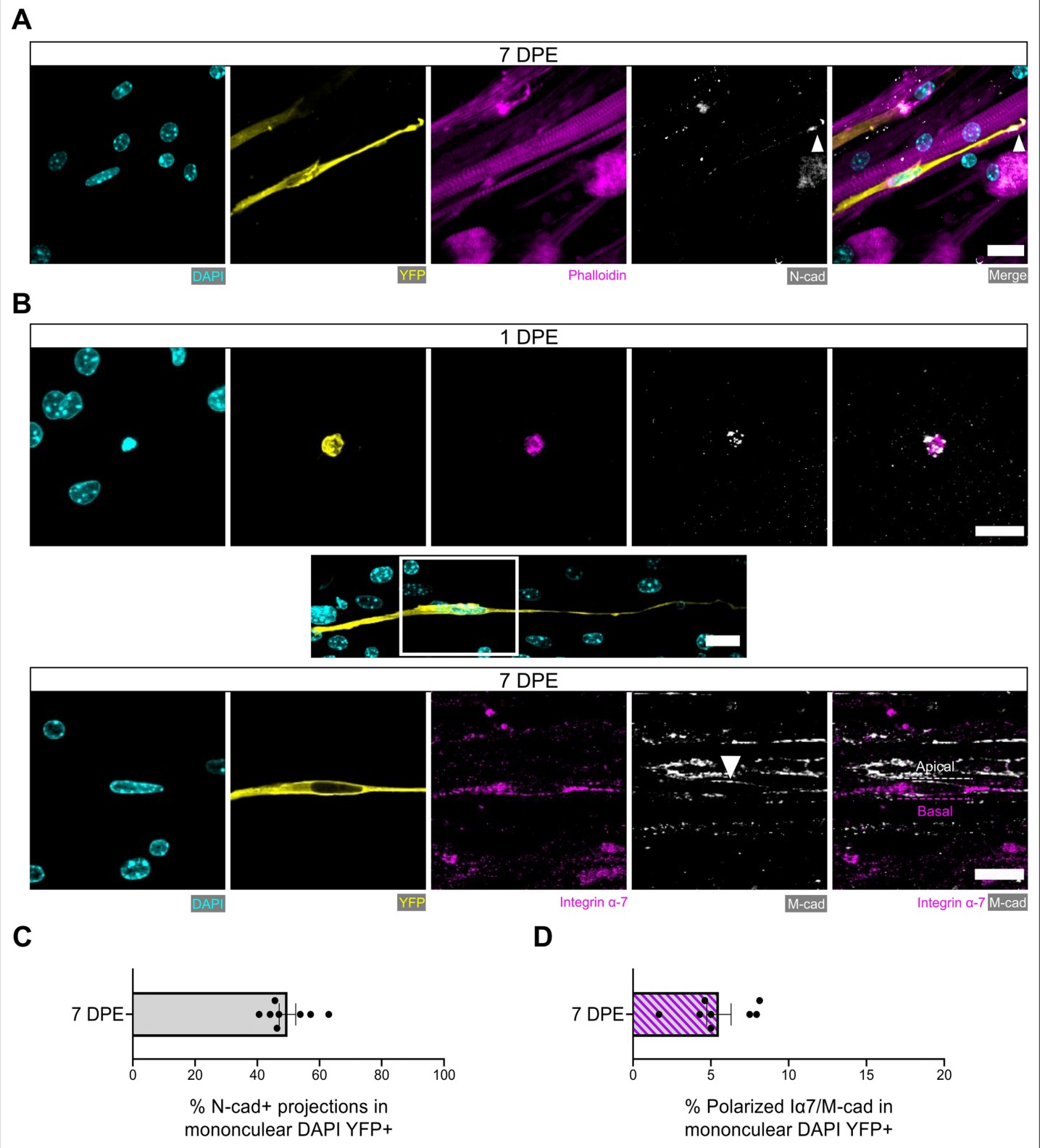

**Figure 6.** Engrafted muscle stem cells (MuSCs) display quiescence and niche-related hallmarks. (**A**) Representative confocal image of a mononuclear donor cell (DAPI: cyan, YFP: yellow) with neighbouring myotubes (Phalloidin: magenta) and N-cadherin (white) localized to the tip of the donor cell projection (white arrowhead). Scale bar, 20 μm. (**B**) Representative confocal images of a mononculear donor cell (DAPI: cyan, YFP: yellow) at 1 day post-engraftment (DPE) (top) and 7 DPE (middle and bottom) expressing integrin α-7 (magenta) and M-cadherin (white). Middle inset image channels

*Figure 6 continued on next page*

*Figure 6 continued*

are separated to produce the bottom images to highlight the polarization of integrin α-7 and M-cadherin (white arrow) to basal and apical orientations, respectively (dotted lines). Scale bars, 20 μm. (**C**) Bar plot showing the percentage of mononuclear DAPI⁺YFP⁺ cells with N-cadherin⁺ cytoplasmic projections at 7 DPE. n=8 across N=3 independent biological replicates. Graph displays mean ± s.e.m. with individual technical replicates. (**D**) Bar plot showing the percentage of mononuclear DAPI⁺YFP⁺ cells with polarized integrin α-7 (Iα7)/M-cadherin expression at 7 DPE. n=8 across N=3 independent biological replicates. Graph displays mean ± s.e.m. with individual technical replicates. Raw data available in *Figure 6—source data 1*.

The online version of this article includes the following source data and figure supplement(s) for figure 6:

**Source data 1.** Raw data for *Figure 6*.

**Figure supplement 1.** Polarized localization of M-cadherin in mononuclear donor cells at 7 days post-engraftment (DPE).

**Figure supplement 2.** Donor cell polarization of niche markers at 7 days post-engraftment (DPE).

period showed a decline in the proportion of c-FOS⁺ (*Figure 7D*) and Ki67⁺ (*Figure 7C*) cells with time, albeit with delayed inactivation kinetics when compared to young MuSCs (*Figure 7C–D*). To further evaluate the quiescent-like state of the engrafted MuSC populations, we quantified the morphology of individual Pax7⁺ donor cells at 7 DPE. On average, aged donor cells had reduced max/min Feret diameter ratio and nuclear eccentricity when compared to young cells at this timepoint, which correlates to the more contracted/rounded morphological characteristics of activated MuSCs (*Figure 7E–G*).

We next pursued a potential rescue of the aged MuSC phenotypes we observed in our engineered cultures. Recent work showed that FoxO transcription factors are responsible for conferring a 'genuine' quiescent state to MuSCs, whereby genetic ablation resulted in a shift towards a 'primed' state (*García-Prat et al., 2020*). Furthermore, FoxO activity was computationally predicted to be regulated by the Igf-Akt pathway, where phosphorylated-Akt causes the phosphorylation of FoxO transcription factors and their translocation to the cytoplasm. Pharmacological inhibition of the AKT pathway using the phosphatidylinositol 3 kinase inhibitor, wortmannin, resulted in increased stemness in primed young MuSCs. Whether this treatment strategy is capable of rescuing aged MuSCs is currently unknown, providing an opportunity to leverage our culture model to explore this hypothesis. First, we confirmed that compared to young MuSCs, aged MuSCs presented increased proliferation and reduced FoxO3a nuclear fluorescent intensity in 2D culture. In this context, wortmannin treatment (10 μM) blunted cell proliferation and increased FoxO3a nuclear localization in both young and aged MuSCs (*Figure 7—figure supplement 1A–C*). Indeed, FoxO3a nuclear fluorescent intensity was comparable between young and aged MuSCs following wortmannin treatment in 2D culture (*Figure 7—figure supplement 1B*).

We then introduced wortmannin to 3D myotube cultures engrafted with young or aged MuSCs. With this treatment, the aged MuSCs maintained a stable Pax7⁺ donor population over time, that was indistinguishable from the untreated young MuSCs cultures (*Figure 7A–B*). Indeed, the proportions of Ki67⁺ and c-FOS⁺ in the aged Pax7⁺ population showed comparable kinetics to the young untreated donor MuSCs (*Figure 7C–D*). As well, the treatment encouraged a greater proportion of young MuSCs to inactivate, and with more rapid kinetics (*Figure 7C*). We also found a trending decrease in donor cell GFP⁺ signal covering tissues at 7 DPE in wortmannin-treated conditions, suggesting reduced differentiation (*Figure 7—figure supplement 2*). Finally, morphological characterization of wortmannin-treated aged MuSCs at 7 DPE showed no change in average max/min Feret diameter ratio, whereas we found a rescue of nuclear eccentricity that matched young MuSCs (*Figure 7E–G*). There were no shifts in the morphological profile of young MuSCs treated with wortmannin (*Figure 7E–G*).

Thus, by introducing aged MuSCs into a young myotube niche we revealed abnormal population maintenance, delays in the inactivation kinetics, and morphological features characteristic of activated MuSCs, which we show can be dampened by modulating AKT signalling in the context of the mini-IDLE assay.

## Discussion

We have developed mini-IDLE, an in vitro functional assay that rapidly induces and sustains murine MuSC inactivation, therein enabling systematic analyses of cellular and molecular mechanisms presiding over the return to quiescence for the first time. MuSCs acquired in vivo-like hallmarks of quiescence in the mini-IDLE assay, that, to our knowledge, have never before been recapitulated in vitro. Through temporal single-cell analyses, we uncovered evidence of population-level adaptations

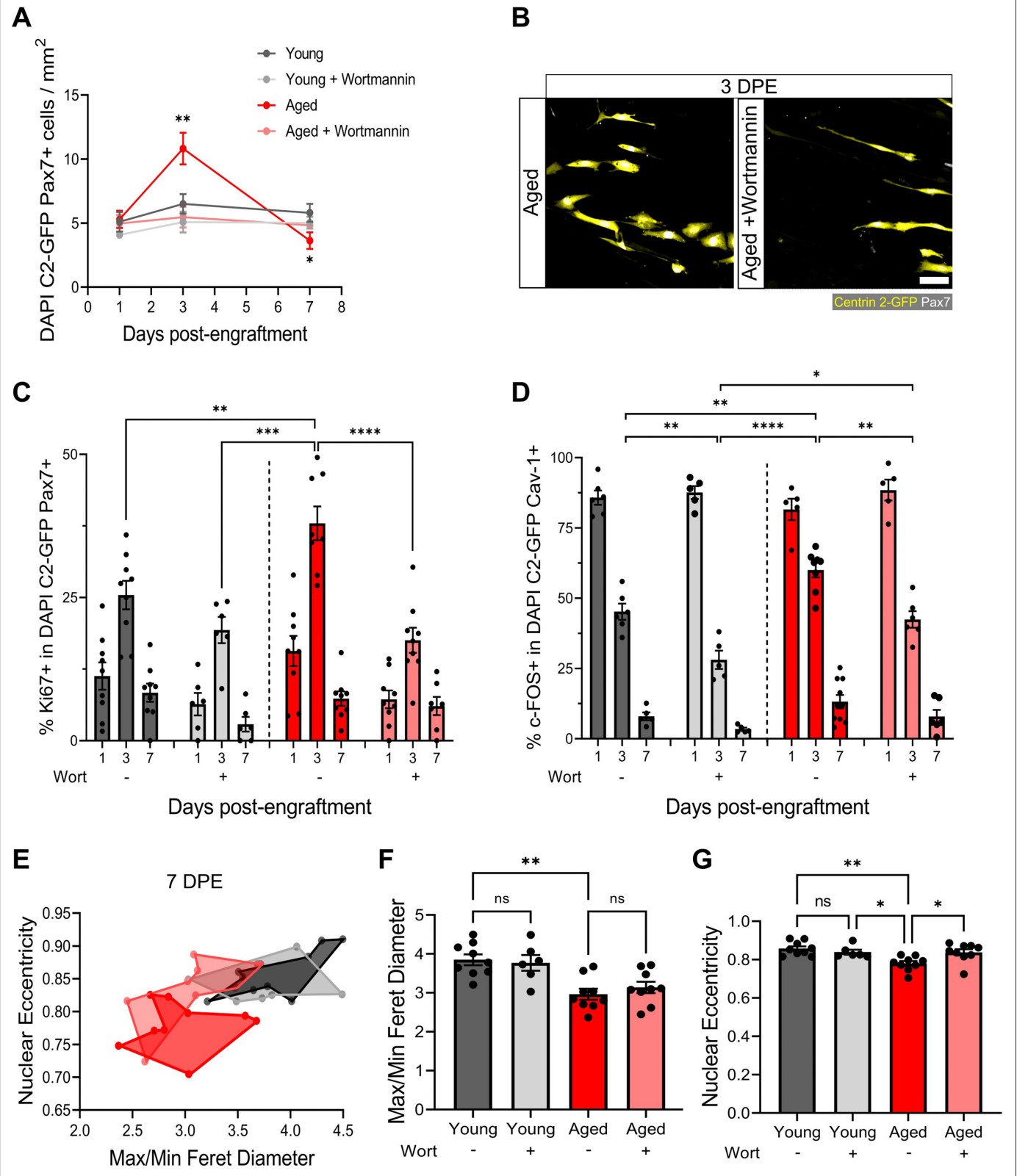

**Figure 7.** Aberrant pool size maintenance and inactivation in aged muscle stem cells (MuSCs) is rescued by wortmannin. (**A**) Quantification of mononuclear DAPI+Centrin 2-GFP (C2-GFP)+Pax7+ cell density per mm² at 1, 3, and 7 days post-engraftment (DPE) between engrafted young and aged MuSCs ± wortmannin (wort) treatment. n=6–9 across N=2–3 independent biological replicates, graph displays mean ± s.e.m.; one-way ANOVA with Dunnet's test for each individual timepoint comparing against the young condition, *p=0.0262, **p=0.0065. (**B**) Representative confocal image of

*Figure 7 continued on next page*

*Figure 7 continued*

donor cells (Centrin 2-GFP:yellow, Pax7:white) from the aged and aged + wortmannin conditions at 3 DPE. Scale bar, 50 μm. (**C**) Bar graph showing the percentage of Ki67$^+$ cells in the DAPI$^+$C2-GFP$^+$Pax7$^+$ mononucleated population at 1, 3, and 7 DPE across experimental conditions (young: dark grey; young + wortmannin: light grey; aged: red; aged + wortmannin: light red). n=6–9 across N=2–3 independent biological replicates, graph displays mean ± s.e.m. with individual technical replicates; one-way ANOVA with Tukey's post-test comparing the conditions against each other at the 3 DPE timepoint, **p=0.0064, ***p=0.0003, ****p<0.0001 (comparisons not shown are ns). (**D**) Bar graph showing the percentage of c-FOS$^+$ cells in the DAPI$^+$C2-GFP$^+$Cav-1$^+$ mononucleated population at 1, 3, and 7 DPE across experimental conditions (young: dark grey; young + wortmannin: light grey; aged: red; aged + wortmannin: light red). n=5–10 across N=2–3 independent biological replicates, graph displays mean ± s.e.m. with individual technical replicates; one-way ANOVA with Tukey's post-test comparing the conditions against each other at the 3 DPE timepoint, *p=0.0169, **p=0.0040, 0.0053, 0.0010, ****p<0.0001 (comparisons not shown are ns). (**E**) Dot graph where each dot represents the average max/min Feret diameter ratio and nuclear eccentricity of the Pax7$^+$ donor cells within the technical replicate (tissue) at the 7 DPE timepoint, colour coded according to experimental condition (young: dark grey; young + wortmannin: light grey; aged: red; aged + wortmannin: light red). (**F**) Bar graph showing the average max/min Feret diameter ratio across experimental conditions, graph displays mean ± s.e.m. with the individual technical replicates from panel E; one-way ANOVA with Tukey's post-test, **p=0.0010 young vs. aged + wortmannin and young + wortmannin vs. aged are also **p=0.0093, 0.0084, but not shown. All other comparisons are not significant. (**G**) Bar graph showing the average nuclear eccentricity across experimental conditions, graph displays mean ± s.e.m. with the individual technical replicates from panel (**E**); one-way ANOVA with Tukey's post-test, *p=0.0402, 0.0216, **p=0.0015. All other comparisons are not significant. Raw data available in *Figure 7—source data 1*.

The online version of this article includes the following source data and figure supplement(s) for figure 7:

**Source data 1.** Raw data for *Figure 7*.

**Figure supplement 1.** Wortmannin treatment blunts cell proliferation and increases FoxO3a nuclear localization in young and aged muscle stem cells (MuSCs) cultured in vitro.

**Figure supplement 1—source data 1.** Raw data for *Figure 7—figure supplement 1*.

**Figure supplement 2.** Wortmannin treatment diminishes GFP coverage at 7 days post-engraftment (DPE).

**Figure supplement 2—source data 1.** Raw data for *Figure 7—figure supplement 2*.

---

to the muscle tissue niche and functionally heterogenous MuSC subpopulations mirroring in vivo heterogeneous activities. We also demonstrate the value proposition of the assay by introducing MuSCs from aged mice and revealing multiple functional deficits tied to an aberrant quiescent state, which we show are partially rescued by wortmannin treatment, a quiescence-reinforcing strategy previously tested on 'primed' MuSCs from young mice (*García-Prat et al., 2020*). To our knowledge, this is the first aging assay 'in a dish' to capture advanced features of aged adult stem cell dysfunction. These breakthroughs, together with the modularity of the assay components, miniaturized format, and validated semi-automated workflows to capture and process phenotypic data, offer an unprecedented opportunity to advance our understanding of MuSC quiescence and regulation in iterated designer niches.

In spite of a stress-induced response to tissue digestion and cell sorting (*Machado et al., 2021*), our data demonstrate that the primary response of most MuSCs introduced to the biomimetic niche is immediate inactivation. A small proportion enter cell cycle prior to inactivation and an even smaller subset directly differentiate and fuse with myotubes in the template. The MuSC population is increasingly regarded as encompassing a continuum of quiescence to activation (*Ancel et al., 2021*), and we believe our assay captures this continuum-influenced functional heterogeneity. More specifically, we expect those MuSCs closer to activation were inclined to differentiate and fuse, while those closer to a deeply quiescent state were resistant to activation cues. Additionally, Pax7$^+$ donor cells at 7 DPE expressing CalcR and/or polarized niche markers represent a little over one-tenth of the population, hinting that a subset of more naive MuSCs are those recapitulating the more 'advanced' hallmarks of quiescence we observed at 7 DPE. This is particularly intriguing taken with studies by others attributing a bona fide stem cell status to a similar proportion of MuSCs within the total population (*Rocheteau et al., 2012*; *Kuang et al., 2007*; *García-Prat et al., 2020*; *Sousa-Victor et al., 2022*). Indeed, taken together with our observation that a subpopulation of engrafted donor MuSCs never enter cell cycle (*Figure 3*), we proport that the biomimetic niche maintains a 'genuine-like' quiescent MuSC population alongside a more 'primed-like' MuSC population, therein offering a tractable culture system with which to identify biochemical and biophysical regulators of these unique states.

Engrafted MuSCs showed population-level control over their response to the niche, which opens up enticing possibilities for studies of MuSC pool size regulation, and to uncover rules dictating niche repopulation. Consistently, increased differentiation was recently recognized as a quality control

mechanism to control MuSC pool size in vivo, biology that is matched in our in vitro studies (*Wang et al., 2022b*). Rules of niche occupancy extoll limits on the number of transplanted MuSCs that can engraft into a recipient muscle (*Arpke et al., 2021*), a barrier that may be broken upon expanding knowledge of MuSC pool size regulators. The data presented also underscores the importance of myotubes encased in a 3D matrix in allowing a persistent Pax7$^+$ pool, and for determining the niche occupancy plateau point, despite a differentiation-inducing culture milieu and the absence of any other cell types. This is perhaps not surprising as many studies tout a role for myofibers in preserving or inducing quiescence, and in controlling MuSC pool size (*Eliazer et al., 2019*; *Sampath et al., 2018*; *Southard et al., 2016*; *Zofkie et al., 2021*). A feedback mechanism between myofibers and MuSCs that is linked to nuclear content is suspected (*Zofkie et al., 2021*); the likes of which could be interrogated in our system.

MuSCs in situ display long cytoplasmic projections that were initially described from electron microscopy analyses (*Schmalbruch, 1978*), and more recently evaluated using tissue clearing and intravital imaging methodologies (*Verma et al., 2018*; *Kann et al., 2022*; *Ma et al., 2022*). The elaborate MuSC morphologies arising in our cultures offer a new opportunity to explore the cellular and molecular mechanisms driving the acquisition of quiescent-like morphologies, but also the relevance of this phenotypic feature on MuSC behaviour and fate. Indeed, long elaborated cytoplasmic projections have been associated with a deeper quiescent state (*Kann et al., 2022*), and have been ruled out as a migratory apparatus (*Ma et al., 2022*), favouring instead a role in 'niche sensing', though that remains to be determined (*Kann et al., 2022*; *Ma et al., 2022*). Quite surprisingly, we found that acquisition of quiescent-like morphologies and anatomical hallmarks was dependent on interactions between MuSCs and their immediate niche, occurring in the absence of other resident muscle cell types. The modular culture assay described herein enables iterative study design and independent molecular perturbations to the niche (myotubes) and the MuSCs to break open knowledge in this area. Indeed, leveraging high-content imaging and CellProfiler workflows for relating morphometric features to fate signatures, we offer proof-of-concept support for the use of morphological features as a non-invasive readout of MuSC quiescence status in our model, thereby facilitating future phenotypic screening efforts.

To date, characterization of functional deficits of aged MuSC populations in vitro have focused on proliferation and colony formation as readouts (*Haroon et al., 2022*; *Cosgrove et al., 2014*; *García-Prat et al., 2013*; *Mouly et al., 2005*; *Schultz and Lipton, 1982*; *Shefer et al., 2006*), and studies of other recognized deficiencies have been restricted to in vivo studies. In recent years, aged MuSC regenerative deficits have been linked to the notion that with age there is a progressive decrease in truly quiescent cells in favour of a greater number of cells in a pre-activated state (*Chakkalakal et al., 2012*; *Kimmel et al., 2020*; *Kimmel et al., 2021*; *Carlson et al., 2008*). Consistent with functional consequences expected of a pre-activated state, we noted aberrant expansion activity at early culture timepoints from a subset of the aged MuSCs seeded within our assay, and a trend towards increased differentiation at later timepoints, that were not observed in young MuSC cultures. Our studies also uncovered delayed inactivation kinetics and also in the acquisition of quiescent-like features. This suggests that a subset of aged MuSCs were unable to properly sense and respond to the pro-quiescent environment. Furthermore, the aged MuSC population was unable to maintain a steady-state pool size, which may imply that MuSC pool regulation is at least partly cell intrinsic and is dependent on both the activation state of MuSCs under steady-state conditions and exposure to activation-inducing cues. Interestingly, the myoblasts used to fabricate all of the muscle tissues for this study were derived from young mice, meaning that the aged MuSCs were exposed to a young niche. We cannot rule out the possibility that the young biomimetic muscle niche partially rescued aged MuSC function, as has been reported by others (*Conboy et al., 2005*; *Lazure et al., 2022*). Indeed, we anticipate that aged MuSCs introduced to muscle tissues fabricated from myoblasts derived from aged donors and/or exposed to an aging systemic environment will induce further functional decline.

Finally, we found that inhibiting Akt signalling restored aged MuSC inactivation kinetics and population control. Our study extends prior work in showing that a strategy demonstrated to confer a genuine quiescent state onto young, activated MuSCs has a similar effect on aged MuSCs. We show that a decline in nuclear FoxOa3 levels is detected in MuSCs at an earlier age than previously thought, and that the nuclear FoxOa3 expression is corrected to youth-like levels by the wortmannin treatment. Wortmannin treatment had only subtle influence on young MuSCs, which may reflect an absence of

stimulatory niche-derived ligands (*García-Prat et al., 2020*). However, we note that the DM culture media contains insulin, and the young and aged MuSCs were each cultured within young muscle tissues.

To conclude, herein we report that mini-IDLE is a culture assay capable of recapitulating aspects of quiescent mouse MuSC biology, in youth and in age, that were previously not possible to study in vitro. By contrast to all other 3D culture systems where the cellular and ECM components are mixed together and introduced at the start of the experiment (*Tiburcy et al., 2019*; *Fleming et al., 2020*; *Rajabian et al., 2021*; *Trevisan et al., 2019*; *Wang et al., 2022a*; *Afshar Bakooshli et al., 2019*; *Madden et al., 2015*; *Afshar et al., 2020*), our method is modular. Amongst the merits of this distinction is the ability to introduce and evenly distribute new cellular components to the assay at any time-point. It is also feasible to genetically modify the MuSC and myoblast components in different ways, and maintain these distinctions, when MuSCs are introduced to the muscle tissue after myotubes have formed. These advantages, together with the simplicity of the approach, assay compatibility with existing semi-automated high-content image acquisition and analysis tools, and high value features of MuSC biology captured by the system, offer a unique opportunity to expand MuSC fundamental knowledge and identify molecular targets to protect MuSC function as animals age.

# Materials and methods
## Animal use protocols and ethics
All animal use protocols were reviewed and approved by the local Animal Care Committee (ACC) within the Division of Comparative Medicine (DCM) at the University of Toronto. All methods in this study were conducted as described in the approved animal use protocols (#20012838) and more broadly in accordance with the guidelines and regulations of the DCM ACC and the Canadian Council on Animal Care. 129-Tg(CAG-EYFP)7AC5Nagy/J (Actin-eYFP) mice (*Hadjantonakis et al., 2002*) were purchased from the Jackson Laboratory by the lab of Dr Derek van der Kooy and shared with our group. Tg:Pax7-nEGFP (i.e. Pax7-nGFP) mice were a gift from Dr Shahragim Tajbakhsh (*Sambasivan et al., 2009*), kindly transferred from the laboratory of Dr Michael Rudnicki at the Ottawa Hospital Research Institute. Unless otherwise indicated, 8- to 12-week-old mice were used for all experiments. A breeding pair of CB6-Tg(CAG-EGFP/CETN2)3-4Jgg/J (Centrin 2-eGFP) transgenic mice

**Table 1.** Cell culture media and solutions.

| Media | Composition |
|---|---|
| FACS Buffer | PBS, 2.5% Goat serum (Gibco, #16210072), 2 mM EDTA (Sigma-Aldrich, #E5134) |
| RBC Lysis Buffer | ddH$_2$O, 0.155 M NH$_4$Cl (Sigma-Aldrich, #A9434), 0.01 M KHCO$_3$ (Sigma-Aldrich, #237205), 0.1 mM EDTA |
| MACS Buffer | PBS, 0.5% Bovine serum albumin (BioShop, #9048-46-8), 2 mM EDTA |
| SAT10 | DMEM/F12 (Gibco, #11320-033), 1% Penicillin-streptomycin (Gibco, #15140-122), 20% Fetal bovine serum (Gibco, 12483-020), 10% Horse serum (Gibco, #16050-122), 1% Glutamax (Gibco, #35050-061), 1% Insulin-transferrin-selenium (Gibco, #41400-045), 1% Non-essential amino acids (Gibco, #11140-050), 1% Sodium pyruvate (Gibco, #11360-070), 50 μM β-mercaptoethanol (Gibco, #21985-023), 5 ng/mL bFGF (ImmunoTools, #11343625) |
| Growth media (GM) | SAT10 – bFGF, 1.5 mg/mL Aminocaproic acid (Sigma-Aldrich, #A2504) |
| Differentiation media (DM) | DMEM (Gibco, #11995-065), 2% Horse serum, 2 mg/mL Aminocarpoic acid, 10 μg/mL Insulin (Sigma, #I6634), 1% Penicillin-streptomycin |
| Blocking Solution | PBS, 10% Goat serum, 0.3% Triton X-100 (BioShop, #TRX777) |
| Physiological Salt Solution (PSS) | 140 mM NaCl (Sigma-Aldrich, #S5886), 5 mM KCl (Sigma-Aldrich, #P3911), 1 mM MgCl$_2$ (Alfa Aesar, #7786-30-3), 10 mM HEPES (BioShop, #7365-45-9), 10 mM Glucose (Sigma-Aldrich, #G8270), 2 mM CaCl$_2$ (Sigma-Aldrich, #C1016), corrected to pH 7.3–7.4 |
| Wash Media | 89% DMEM, 10% Fetal bovine serum, 1% Penicillin-streptomycin |

(*Higginbotham et al., 2004*) were kindly shared by Jeffrey Martens (University of Florida), and maintained by breeding for use in the aging studies. Young mice were between 4 and 5 months and aged mice between 24 and 26 months of age.

## MACS of primary mouse MuSCs

Primary mouse MuSCs were isolated from mouse hindlimb muscle using a modified method previously reported by our group (*Davoudi et al., 2022*). Briefly, ≈1 g of muscle tissues was dissected from the hindlimb muscles of a humanely euthanized mouse and placed into a GentleMACS dissociation tube (Miltenyi Biotec, #130-096-334). Seven mL of DMEM (Gibco, #11995-073) with 630 U/mL Type 1A collagenase from *Clostridium histolyticum* (Sigma, #C9891) was added to the tube, and the sample was physically dissociated using a GentleMACS dissociator (Miltenyi Biotec, #130-096-334) using the 'skeletal muscle' setting. The tube was then placed on an orbital shaker in a 37°C incubator for 1 hr. The digested tissue was triturated 10 times through a 10 mL pipette, after which an additional 440 U of Type 1A collagenase was added along with Dispase II (Life Technologies, #17105041) and DNAse I (Bio Basic, #9003-98-9) at a final concentration of 0.04 U/mL and 100 µg/mL, respectively. The tube was again placed on an orbital shaker in a 37°C incubator for 1 hr. The sample was then slowly passed through a 20 G needle 15 times and then resuspended in 7 mL of FACS buffer (*Table 1*). The solution was passed through a 70 µm cell strainer (Miltenyi Biotec, #130-098-462) followed by a 40 µm cell strainer (Corning, #352340). The filtered mixture was then centrifuged at 400 × *g* for 15 min and the supernatant aspirated. The pellet was resuspended in 1 mL of 1× red blood cell (RBC) lysis buffer (*Table 1*) and then incubated at room temperature (RT) for 8 min. Nine mL of FACS buffer was added to the tube and the mixture was centrifuged at 400 × *g* for 15 min followed by supernatant aspiration.

The cell pellet was then incubated in a 4°C fridge with rocking for 15 min in 100 µL of MACS buffer and 25 µL of lineage depletion microbeads from the Satellite Cell Isolation Kit (Miltenyi Biotec, #130-104-268) according to the manufacturer's instructions. Another 375 µL of MACS buffer was then added, and the lineage positive cells depleted by flowing the solution, by gravity, through an LS column in a magnetic field (Miltenyi Biotec, #130-042-401) (*Table 1*). The resulting flow through was collected, corrected to 5 mL and then centrifuged at 400 × *g* for 5 min. The pellet was then subjected to a second round of lineage depletion using a fresh LS column in a magnetic field. The flow through was corrected to 5 mL, centrifuged, followed by supernatant aspiration, and then the cell pellet was resuspended in 100 µL of MACS buffer and 25 µL of anti-integrin α7 microbeads (Miltenyi Biotec, #130-104-261) for incubation at 4°C for 15 min; 375 µL MACS buffer was added, and the integrin α-7$^+$ was enriched by running the solution through a third LS column in a magnetic field. In this instance, the flow through was discarded, the column was removed from the magnetic field, and then flushed with 5 mL of MACS buffer which was collected in a 15 mL conical tube. The tube was spun to generate a cell pellet enriched for integrin α-7$^+$ MuSCs. To establish and validate the protocol, which differs from the manufacturer's protocol by the introduction of extra lineage depletion steps, α-7$^+$ MuSCs were isolated from Pax7-nGFP transgenic mice. In these experiments the cell pellet was resuspended in 0.5 mL FACS buffer and incubated with DRAQ5 for 15 min at RT. After 3×5 min FACS buffer washes and centrifuge spins, the pellet was resuspended in 0.5 mL of FACS buffer and propidium iodide (PI) was added to the tube. The resuspended cells were then evaluated using the Accuri C6 Flow Cytometer (BD Biosciences) whereby we collected 30,000 events. The DRAQ5$^+$Pax-nGFP$^+$PI$^-$ cell population was quantified from the flow cytometric data using FlowJo V10 software.

## Primary mouse myoblast derivation and maintenance

Primary mouse myoblasts were derived from freshly MACS enriched integrin α-7$^+$ MuSC populations. One day before cell plating, culture dishes were coated at 4°C overnight with collagen I at a 1:8 concentration diluted in ddH$_2$O (Gibco, #A10483-01). The next day, excess collagen I solution was removed, and the dish culture surfaces were dried at RT for 15–20 min followed by a PBS wash prior to use. Immediately after MACS isolation, lineage depleted integrin α-7$^+$ enriched MuSCs were resuspended in SAT10 media (*Table 1*) and plated into collagen-coated dishes. A full media change was performed 48 hr after plating with half media changes every 2 days thereafter. Cells were grown to 70% confluency and passaged at least five times to produce a primary mouse myoblast line, and then used from passage 5–9 for experiments.

## Murine myotube template fabrication and MuSC seeding

One day prior to seeding myotube templates, black 96-well clear bottom plates (PerkinElmer, #6055300) were coated with 5% pluronic acid (Sigma-Aldrich, #P2443) and incubated overnight at 4°C. The next day, excess pluronic solution was removed, and plates were left at RT for 15–20 min to dry well surfaces. Cellulose paper (MiniMinit) was cut into 5 mm discs using a biopsy punch (Integra, #MLT3335), autoclaved, and then placed into pluronic acid-coated wells of the 96-well plate. A stock thrombin solution (100 U/mL, Sigma-Aldrich, #T6884) was then diluted to 0.8 U/mL in PBS, and then 4 µL was diffused into the paper discs and left to dry at RT. Meanwhile, a 10 mg/mL fibrinogen solution was made by dissolving lyophilized fibrinogen (Sigma-Aldrich, #F8630) in a 0.9 % wt/vol solution of NaCl (Sigma-Aldrich, #S5886) and then filtered through a 0.22 µm syringe filter (Sarstedt, #83.1826.001). Primary myoblasts were then trypsinized, counted using a hemacytometer, and then resuspended in an ECM-mimicking slurry comprised of 40% DMEM, 40% Fibrinogen, and 20% Geltrex (Thermo Fisher Scientific, #A1413202) at a concentration of 25,000 cells per 4 µL. The cell/ECM solution was then diffused into dry thrombin-containing paper discs and left to gel at 37°C for 5 min. Two-hundred µL GM (*Table 1*) was introduced to each hydrogel containing culture well and plates were returned to a cell culture incubator (37°C, 5% $CO_2$) for 2 days (day –2 to 0). On day 0 of differentiation, a full media change was conducted to transition cultures to DM (*Table 1*). Half media changes with DM were performed every other day from thereafter.

Unless otherwise indicated, on day 5 of myotube template culture integrin α-7+ MuSCs were prospectively isolated and resuspended in SAT10 media replete of FGF2. After visual inspection to confirm uniform distribution of myotubes across the template, myotube templates that passed quality control (>95% of templates) were carefully removed from the 96-well plate using tweezers and placed in an ethanol-sterilized plastic container containing long strips of polydimethylsiloxane sitting on top of a moist paper towel. Quickly, 4 µL of the resuspended MuSC solution containing the desired number of MuSCs was placed onto each tissue and evenly spread over the tissue surface using a cell spreader. The plastic container was then sealed with a tight fitting lid and placed in the 37°C incubator for 1 hr before putting the tissues back into their wells using tweezers. For ROCK inhibition studies, Y-27632 (Tocris, #1254, dissolved in water) was added to the culture media at 50 µM and refreshed every other day, with water serving as the vehicle control. For aged MuSC-related studies, Dr Louise A. Moyle (Toronto, Canada) kindly harvested, minced, and cryopreserved hindlimb muscle from young or litter-matched aged Centrin 2-eGFP mice (*Higginbotham et al., 2004*) that were later thawed and underwent the MACS protocol detailed above. The MuSCs were then resuspended in SAT10 media replete of FGF2 but with added wortmannin (10 µM, Sigma-Aldrich, #W1628) or a dimethyl sulfoxide (DMSO) control (Sigma-Aldrich, #D8418). Four µL of the resuspended MuSCs containing ≈500 cells were subsequently seeded onto individual tissues. After 1 hr, tissues were put back into their wells. The wortmannin (or DMSO) was then added to the culture media (also at 10 µM) and refreshed every other day during media changes.

## Tissue fixation and immunolabelling

At the indicated tissue endpoints, samples were quickly washed 3× with PBS before fixation with 100 µL of 4% paraformaldehyde (PFA, Thermo Fisher Scientific, #50980494) for 12 min at RT. After 3×10 min washes with cold PBS (4°C), blocking and permeabilization was performed using 100 µL of blocking solution (*Table 1*) for 30 min at RT. Afterwards, primary antibodies were diluted in blocking solution as indicated in *Table 2* and 50 µL was added to each tissue and incubated overnight at 4°C. After 3×10 min washes with cold PBS, tissues were incubated for 45 min at RT in 50 µL of secondary antibodies and molecular probes diluted in blocking solution (see *Table 2*), followed by 3×10 min washes with cold PBS. A limitation of the cellulose papers is that they cast autofluorescence in the blue channel, which can give off intense background noise. Therefore, for nuclei detection, DAPI was sometimes used as the signal intensity was generally high enough to allow thresholding of paper fibers out of confocal images. Batch to batch differences in DAPI, or in cases when tissues become dry during staining, can result in DAPI images where the cellulose fibers are visualized, although even in these cases the nuclei can still be clearly discerned.

## Image acquisition

Confocal imaging was performed using the Perkin-Elmer Operetta CLS High-Content Analysis System and the associated Harmony software. Prior inserting the 96-well plate into the Operetta, the PBS

**Table 2.** Antibodies.

| Antibody | Host Species | Dilution | Manufacturer |
| --- | --- | --- | --- |
| DAPI | – | 1:1000 | Roche, #10236276001 |
| Phalloidin 568 | – | 1:400 | Life Technologies, #A12380 |
| Propidium iodide | – | 1:1000 | Sigma-Aldrich, #P4863 |
| DRAQ5 | – | 1:800 | Cell Signaling Technology, #4084L |
| Anti-sarcomeric α-actinin | Mouse | 1:800 | Sigma-Aldrich, #A7811 |
| Anti-GFP | Chicken | 1:500 | Abcam, #ab13970 |
| Anti-Pax7 | Mouse IgG1 | 1.5:1 | In-house supernatant from hybridoma cell line (DSHB) |
| Anti-caveolin-1 | Rabbit | 1:300 | Abcam, #ab2910 |
| Anti-c-FOS | Mouse IgG1 | 1:250 | Santa-Cruz, #sc-166940 |
| Anti-Ki67 | Rabbit | 1:300 | Abcam, #ab16667 |
| Anti-N-cadherin | Mouse IgG1 | 1:250 | Santa-Cruz, #sc-8424 |
| Anti-MyoD | Mouse IgG2b | 1:300 | Santa-Cruz, #sc-377460 |
| Anti-CalcR | Rabbit | 1:250 | Abcam, #ab11042 |
| Anti-DDX6 | Rabbit | 1:400 | Cederlane Labs, #A300-461A-T |
| Anti-Integrin α-7 | Rabbit | 1:250 | Abcam, #ab203254 |
| Anti-M-cadherin | Mouse IgG1 | 1:250 | Santa-Cruz, #sc-81471 |
| Anti-Laminin α-2 | Rat | 1:400 | Abcam, #ab11576 |
| Anti-FoxO3a | Mouse IgG1 | 1:250 | Santa-Cruz, #sc-48348 |
| Alexa Fluor 488 Anti-mouse IgG (H+L) | Goat | 1:500 | Invitrogen, #A11001 |
| Alexa Fluor 488 Anti-chicken IgGY (H+L) | Goat | 1:500 | Invitrogen, #A11039 |
| Alexa Fluor 546 Anti-mouse IgG (H+L) | Goat | 1:250 | Invitrogen, #A11003 |
| Alexa Fluor 546 Anti-rabbit IgG (H+L) | Goat | 1:250 | Invitrogen, #A11010 |
| Alexa Fluor 546 Anti-rat IgG (H+L) | Goat | 1:400 | Invitrogen, #A11081 |
| Alexa Fluor 546 Anti-mouse IgG2b | Goat | 1:300 | Invitrogen, #A21141 |
| Alexa Fluor 555 picolyl azide | – | 1.2:500 | Invitrogen, #C10638B |
| Alexa Fluor 647 Anti-mouse IgG1 | Goat | 1:250 | Invitrogen, #A21240 |
| Alexa Fluor 647 Anti-rabbit IgG (H+L) | Goat | 1:250 | Life Technologies, #A21245 |

was removed from the wells of the plate to prevent tissues from shifting during imaging, and they were carefully positioned in the middle of the wells using tweezers. For stitched pictures, images were collected using the 10× air objective (Two Peak autofocus, NA 1.0 and Binning of 1). For MuSC analysis, images were collected using the 20× and 40× water immersion objectives (Two Peak autofocus, NA 1.0 and 1.1, and Binning of 1). All images were exported off the Harmony software in their raw form. Subsequent stitching, max projections, etc. was performed using the ImageJ-BIOP Operetta Import Plugin available on c4science (*ijs-Perkin Elmer Operetta CLS, Stitching And Export, 2022*). For imaging of MuSC niche markers, the Olympus FV-1000 confocal microscope and Olympus

**Table 3.** Experimental replicate breakdown and statistical analysis.

| Figure | Independent technical and biological replicates (n, N) | Images per technical replicate (tissue) | n to calculate statistics/error bars | Statistical test |
|---|---|---|---|---|
| 1D | **SAA coverage** Day 2: n=12 across N=4 Day 5: n=12 across N=4 Day 10: n=15 across N=5 Day 14: n=15 across N=5 Day 16: n=12 across N=4 Day 18: n=12 across N=4 **Fusion index** Day 2: n=9 across N=3 Day 5: n=12 across N=4 Day 10: n=18 across N=6 Day 14: n=15 N=5 Day 16: n=6 across N=2 Day 18: n=12 across N=4 | **SAA coverage:** 21 images stitched together **Fusion Index:** 9 | **SAA coverage** Day 2: n=12 Day 5: n=12 Day 10: n=15 Day 14: n=15 Day 16: n=12 Day 18: n=12 **Fusion index** Day 2: n=9 Day 5: n=12 Day 10: n=18 Day 14: n=15 Day 16: n=6 Day 18: n=12 | One-way ANOVA with Tukey's post-test |
| 1E | Day 2: n=12 across N=4 Day 5: n=12 across N=4 Day 10: n=9 across N=3 Day 14: n=12 across N=4 Day 16: n=12 across N=4 Day 18: n=9 across N=3 | 3 reads | Day 2: n=12 Day 5: n=12 Day 10: n=9 Day 14: n=12 Day 16: n=12 Day 18: n=9 | One-way ANOVA with Tukey's post-test |
| 2D | **200 MuSCs** 1DPE: n=11 across N=4 3DPE: n=12 across N=4 7DPE: n=12 across N=4 **500 MuSCs** 1DPE: n=8 across N=3 3DPE: n=8 across N=3 7DPE: n=9 across N=3 **1500 MuSCs** 1DPE: n=7 across N=3 3DPE: n=9 across N=3 7DPE: n=11 across N=4 **2500 MuSCs** 1DPE: n=9 across N=3 3PE: n=8 across N=3 7DPE: n=9 across N=3 | 25 | **200 MuSCs** 1DPE: n=11 3DPE: n=12 7DPE: n=12 **500 MuSCs** 1DPE: n=8 3DPE: n=8 7DPE: n=9 **1500 MuSCs** 1DPE: n=7 3DPE: n=9 7DPE: n=11 **2500 MuSCs** 1DPE: n=9 3PE: n=8 7DPE: n=9 | One-way ANOVA with Dunnet's test for each individual timepoint comparing against the 500 MuSC condition |
| 3B | 1DPE: n=9 across N=3 3DPE: n=9 across N=3 7DPE: n=9 across N=3 | 25 | 1DPE: n=9 3DPE: n=9 7DPE: n=9 | One-way ANOVA with Tukey's post-test comparing the FOS- proportions of each timepoint |
| 3C | 1DPE: n=10 across N=3 3DPE: n=11 across N=4 7DPE: n=11 across N=4 | 25 | 1DPE: n=10 3DPE: n=11 7DPE: n=11 | One-way ANOVA with Tukey's post-test comparing the Ki67- proportions of each timepoint |
| 3E | n=15 across N=5 | 25 | n=15 | – |
| 3G | **PSS** n=16 across N=5 **2.4% BaCl** n=18 across N=6 | 25 | **PSS** n=16 **2.4% BaCl** n=18 | Unpaired t-test of the Ki67- proportions of both conditions |
| 4B | 1DPE: n=9 across N=3 3DPE: n=10 across N=3 7DPE: n=9 across N=3 | 25 | 1DPE: n=9 3DPE: n=10 7DPE: n=9 | – |
| 4C | 1DPE: n=9 across N=3 3DPE: n=9 across N=3 7DPE: n=9 across N=3 | 25 | 1DPE: n=9 3DPE: n=9 7DPE: n=9 | – |
| 4D | 1DPE: n=6 across N=2 3DPE: n=6 across N=2 7DPE: n=6 across N=2 | 25 | 1DPE: n=6 3DPE: n=6 7DPE: n=6 | – |

*Table 3 continued on next page*

*Table 3 continued*

| Figure | Independent technical and biological replicates (n, N) | Images per technical replicate (tissue) | n to calculate statistics/error bars | Statistical test |
|---|---|---|---|---|
| 4E | 1DPE: n=6 across N=2<br>3DPE: n=6 across N=2<br>7DPE: n=6 across N=2 | 25 | 1DPE: n=6 3DPE: n=6 7DPE: n=6 | – |
| 4F | 1DPE: n=15 across N=4<br>3DPE: n=14 across N=4<br>7DPE: n=13 across N=4 | 25 | 1DPE: n=15 3DPE: n=14 7DPE: n=13 | – |
| 5C | 1DPE: n=916 across N=4<br>3DPE: n=980 across N=4<br>7DPE: n=737 across N=3 | 25 | – | – |
| 5E | 1DPE: n=639 across N=3<br>3DPE: n=770 across N=3<br>7DPE: n=676 across N=3 | 25 | 1DPE: n=639 3DPE: n=770 7DPE: n=676 | One-way ANOVA with Tukey's post-test |
| 5F | n=676 across N=3 | 25 | Bin 2=147 Bin 3=135 Bin 4=89 Bin 5=69 Bin 6=66 Bin 7=48 Bin 8=44 Bin 9=30 Bin 9+=48 | One-way ANOVA with test for linear trend |
| 6C | n=8 across N=3 | 25 | – | – |
| 6D | n=8 across N=3 | 45 | – | – |
| 7A | **Young** 1DPE: n=9 across N=3 3DPE: n=9 across N=3 7DPE: n=9 across N=3 **Young + wortmannin** 1DPE: n=6 across N=2 3DPE: n=6 across N=2 7DPE: n=6 across N=2 **Aged** 1DPE: n=9 across N=3 3DPE: n=8 across N=3 7DPE: n=9 across N=3 **Aged + wortmannin** 1DPE: n=9 across N=3 3DPE: n=9 across N=3 7DPE: n=7 across N=3 | 25 | **Young** 1DPE: n=9 3DPE: n=9 7DPE: n=9 **Young + wortmannin** 1DPE: n=6 3DPE: n=6 7DPE: n=6 **Aged** 1DPE: n=9 3DPE: n=8 7DPE: n=9 **Aged + wortmannin** 1DPE: n=9 3DPE: n=9 7DPE: n=7 | One-way ANOVA with Dunnet's test for each individual timepoint comparing against the Young condition |
| 7C | **Young** 1DPE: n=9 across N=3 3DPE: n=9 across N=3 7DPE: n=9 across N=3 **Young + wortmannin** 1DPE: n=6 across N=2 3DPE: n=6 across N=2 7DPE: n=6 across N=2 **Aged** 1DPE: n=9 across N=3 3DPE: n=8 across N=3 7DPE: n=9 across N=3 **Aged + wortmannin** 1DPE: n=9 across N=3 3DPE: n=9 across N=3 7DPE: n=7 across N=3 | 25 | **Young** 1DPE: n=9 3DPE: n=9 7DPE: n=9 **Young + wortmannin** 1DPE: n=6 3DPE: n=6 7DPE: n=6 **Aged** 1DPE: n=9 3DPE: n=8 7DPE: n=9 **Aged + wortmannin** 1DPE: n=9 3DPE: n=9 7DPE: n=7 | One-way ANOVA with Tukey's post-test comparing the conditions against each other at the 3 DPE timepoint |

*Table 3 continued on next page*

*Table 3 continued*

| Figure | Independent technical and biological replicates (n, N) | Images per technical replicate (tissue) | n to calculate statistics/error bars | Statistical test |
|---|---|---|---|---|
| 7D | **Young** 1DPE: n=6 across N=2 3DPE: n=6 across N=2 7DPE: n=5 across N=2 **Young + wortmannin** 1DPE: n=5 across N=2 3DPE: n=5 across N=2 7DPE: n=5 across N=2 **Aged** 1DPE: n=5 across N=2 3DPE: n=8 across N=3 7DPE: n=10 across N=3 **Aged + wortmannin** 1DPE: n=5 across N=2 3DPE: n=6 across N=2 7DPE: n=6 across N=2 | 25 | **Young** 1DPE: n=6 3DPE: n=6 7DPE: n=5 **Young + wortmannin** 1DPE: n=5 3DPE: n=5 7DPE: n=5 **Aged** 1DPE: n=5 3DPE: n=8 7DPE: n=10 **Aged + wortmannin** 1DPE: n=5 3DPE: n=6 7DPE: n=6 | One-way ANOVA with Tukey's post-test comparing the conditions against each other at the 3 DPE timepoint |
| 7E | **Young** n=9 across N=3 **Young + wortmannin** n=6 across N=2 **Aged** n=9 across N=3 **Aged + wortmannin** n=9 across N=3 | 25 | – | – |
| 7F | **Young** n=9 across N=3 **Young + wortmannin** n=6 across N=2 **Aged** n=9 across N=3 **Aged + wortmannin** n=9 across N=3 | 25 | **Young** n=9 **Young + wortmannin** n=6 **Aged** n=9 **Aged + wortmannin** n=9 | One-way ANOVA with Tukey's post-test |
| 7G | **Young** n=9 across N=3 **Young + wortmannin** n=6 across N=2 **Aged** n=9 across N=3 **Aged + wortmannin** n=9 across N=3 | 25 | **Young** n=9 **Young + wortmannin** n=6 **Aged** n=9 **Aged + wortmannin** n=9 | One-way ANOVA with Tukey's post-test |
| F1-S1B | **10,000** n=12 across N=4 **25,000** n=12 across N=4 **50,000** n=12 across N=4 | 21 images stitched together | **10,000** n=12 **25,000** n=12 **50,000** n=12 | One-way ANOVA with Tukey's post-test |
| F1-S2C | **3D** 1DPE: n=6 across N=3 3DPE: n=9 across N=3 7DPE: n=9 across N=3 **2 D** 1DPE: n=9 across N=3 3DPE: n=9 across N=3 7DPE: n=9 across N=3 | 21 images stitched together | **3D** 1DPE: n=6 3DPE: n=9 7DPE: n=9 **2D** 1DPE: n=9 3DPE: n=9 7DPE: n=9 | One-way ANOVA with Tukey's post-test |
| F1-S2D | **3D** 1DPE: n=6 across N=3 3DPE: n=9 across N=3 7DPE: n=9 across N=3 **2 D** 1DPE: n=9 across N=3 3DPE: n=9 across N=3 7DPE: n=9 across N=3 | 25 | **3D** 1DPE: n=6 3DPE: n=9 7DPE: n=9 **2D** 1DPE: n=9 3DPE: n=9 7DPE: n=9 | One-way ANOVA with Tukey's post-test |
| F1-S2E | **3D** 1DPE: n=6 across N=3 3DPE: n=9 across N=3 7DPE: n=9 across N=3 **2 D** 1DPE: n=9 across N=3 3DPE: n=9 across N=3 7DPE: n=9 across N=3 | 2 | **3D** 1DPE: n=6 3DPE: n=9 7DPE: n=9 **2D** 1DPE: n=9 3DPE: n=9 7DPE: n=9 | One-way ANOVA with Tukey's post-test |

*Table 3 continued on next page*

*Table 3 continued*

| Figure | Independent technical and biological replicates (n, N) | Images per technical replicate (tissue) | n to calculate statistics/error bars | Statistical test |
|---|---|---|---|---|
| F1-S2F | **3D** 1DPE: n=6 across N=3 3DPE: n=9 across N=3 7DPE: n=9 across N=3 **2 D** 1DPE: n=9 across N=3 3DPE: n=9 across N=3 7DPE: n=9 across N=3 | 2 | **3D** 1DPE: n=6 3DPE: n=9 7DPE: n=9 **2D** 1DPE: n=9 3DPE: n=9 7DPE: n=9 | One-way ANOVA with Tukey's post-test |
| F1-S2G | **3D** 1DPE: n=6 across N=3 3DPE: n=9 across N=3 7DPE: n=9 across N=3 **2 D** 1DPE: n=9 across N=3 3DPE: n=9 across N=3 7DPE: n=9 across N=3 | 2 | **3D** 1DPE: n=6 3DPE: n=9 7DPE: n=9 **2D** 1DPE: n=9 3DPE: n=9 7DPE: n=9 | One-way ANOVA with Tukey's post-test |
| F2-S1C | n=11 across N=4 | 25 | n=11 | – |
| F3-S1B | 1DPE: n=8 across N=3 3DPE: n=7 across N=3 7DPE: n=15 across N=5 | 25 | 1DPE: n=8 3DPE: n=7 7DPE: n=15 | One-way ANOVA with Tukey's post-test |
| F3-S2A | **BI** n=8 across N=3 **PSS** n=7 across N=3 **2.4%** BaCl n=9 across N=3 | 21 images stitched together | **BI** n=8 **PSS** n=7 **2.4%** BaCl n=9 | One-way ANOVA with Tukey's post-test |
| F3-S2B | **BI** n=7 across N=3 **PSS** n=9 across N=3 **2.4%** BaCl n=9 across N=3 | 25 | **BI** n=7 **PSS** n=9 **2.4%** BaCl n=9 | One-way ANOVA with Tukey's post-test |
| F3-S3C | **200 MuSCs** n=11 across N=4 **500 MuSCs** n=15 across N=5 **1500 MuSCs** n=16 across N=5 **2500 MuSCs** n=13 across N=4 | 25 | **200 MuSCs** n=11 **500 MuSCs** n=15 **1500 MuSCs** n=16 **2500 MuSCs** n=13 | One-way ANOVA with Tukey's post-test |
| F3-S3D | **200 MuSCs** n=12 across N=4 **500 MuSCs** n=9 across N=3 **1500 MuSCs** n=12 across N=4 **2500 MuSCs** n=9 across N=4 | 25 | **200 MuSCs** n=12 **500 MuSCs** n=9 **1500 MuSCs** n=12 **2500 MuSCs** n=9 | One-way ANOVA with Tukey's post-test |
| F4-S1B | **Day 5** n=15 across N=5 **Day 0** n=11 across N=4 | 25 | **Day 5** n=15 **5 Day 0** n=11 | Unpaired t-test |
| F5-S1B | **2D** n=9 across N=3 **2D+myotubes** n=8 across N=3 **3D+myotubes** n=14 across n=3 | 25 | **2D** n=9 **2D+myotubes** n=8 **3D+myotubes** n=14 | One-way ANOVA with Tukey's post-test |
| F5-S1C | **2D** n=9 across N=3 **2D+myotubes** n=8 across N=3 **3D+myotubes** n=14 across n=3 | 25 | **2D** n=9 **2D+myotubes** n=8 **3D+myotubes** n=14 | One-way ANOVA with Tukey's post-test |
| F5-S2C | 1DPE: n=35 across N=3 3DPE: n=45 across N=3 7DPE: n=45 across N=3 | Every 5 images is from 1 tissue | n=125 | Simple linear regression |
| F5-S3A | 1DPE: n=916 across N=4 3DPE: n=980 across N=4 7DPE: n=737 across N=3 | 25 | 1DPE: n=916 3DPE: n=980 7DPE: n=737 | – |

*Table 3 continued on next page*

*Table 3 continued*

| Figure | Independent technical and biological replicates (n, N) | Images per technical replicate (tissue) | n to calculate statistics/error bars | Statistical test |
|---|---|---|---|---|
| F5-S3C | Pax7+/MyoD- 3DPE: n=12 across N=4 7DPE: n=11 across N=3 Pax7+/MyoD+ 3DPE: n=11 across N=4 7DPE: n=10 across N=3 | 25 | Pax7+/MyoD- 3DPE: n=12 7DPE: n=11 Pax7+/MyoD+ 3DPE: n=11 7DPE: n=10 | Unpaired t-tests |
| F5-S4B | Control 1DPE: n=7 across N=3 3DPE: n=8 across N=3 7DPE: n=9 across N=3 Y-27632 1DPE: n=9 across N=3 3DPE: n=9 across N=3 7DPE: n=9 across N=3 | 25 | Control 1DPE: n=7 3DPE: n=8 7DPE: n=9 Y-27632 1DPE: n=9 3DPE: n=9 7DPE: n=9 | Unpaired t-tests for each individual timepoint |
| F5-S4C | Control 1DPE: n=6 across N=2 3DPE: n=6 across N=2 7DPE: n=6 across N=2 Y-27632 1DPE: n=6 across N=2 3DPE: n=6 across N=2 7DPE: n=6 across N=2 | 25 | Control 1DPE: n=6 3DPE: n=6 7DPE: n=6 Y-27632 1DPE: n=6 3DPE: n=6 7DPE: n=6 | Unpaired t-tests for each individual timepoint |
| F5-S4D | Control 1DPE: n=7 across N=3 3DPE: n=8 across N=3 7DPE: n=9 across N=3 Y-27632 1DPE: n=9 across N=3 3DPE: n=9 across N=3 7DPE: n=9 across N=3 | 25 | Control 1DPE: n=7 3DPE: n=8 7DPE: n=9 Y-27632 1DPE: n=9 3DPE: n=9 7DPE: n=9 | Unpaired t-tests for each individual timepoint |
| F5-S5A | Control 1DPE: n=666 across N=3 3DPE: n=994 across N=3 7DPE: n=783 across N=3 Y-27632 1DPE: n=1,255 across N=3 3DPE: n=1,158 across N=3 7DPE: n=598 across N=3 | 25 | Control 1DPE: n=666 3DPE: n=994 7DPE: n=783 Y-27632 1DPE: n=1255 3DPE: n=1158 7DPE: n=598 | – |
| F5-S5B | Pax7+/MyoD- and Pax7+/MyoD+(Control 1DPE: n=7 across N=3 3DPE: n=8 across N=3 7DPE: n=9 across N=3 Y-27632 1DPE: n=9 across N=3 3DPE: n=9 across N=3 7DPE: n=9 across N=3) | 25 | Pax7+/MyoD- and Pax7+/MyoD+ (Control 1DPE: n=7 3DPE: n=8 7DPE: n=9 Y-27632 1DPE: n=9 3DPE: n=9 7DPE: n=9) | One-way ANOVA with Tukey's post-test within each individual timepoints |
| F5-S5C | Pax7+/MyoD- and Pax7+/MyoD+(Control 1DPE: n=7 across N=3 3DPE: n=8 across N=3 7DPE: n=9 across N=3 Y-27632 1DPE: n=9 across N=3 3DPE: n=9 across N=3 7DPE: n=9 across N=3) | 25 | Pax7+/MyoD- and Pax7+/MyoD+ (Control 1DPE: n=7 3DPE: n=8 7DPE: n=9 Y-27632 1DPE: n=9 3DPE: n=9 7DPE: n=9) | One-way ANOVA with Tukey's post-test within each individual timepoints |
| F5-S5E | Pax7+/MyoD- and Pax7+/MyoD+(Control 1DPE: n=7 across N=3 3DPE: n=8 across N=3 7DPE: n=9 across N=3 Y-27632 1DPE: n=9 across N=3 3DPE: n=9 across N=3 7DPE: n=9 across N=3) | 25 | Control 1DPE: n=7 3DPE: n=8 7DPE: n=9 Y-27632 1DPE: n=9 3DPE: n=9 7DPE: n=9 | One-way ANOVA with Tukey's post-test within each individual timepoints |

*Table 3 continued*

| Figure | Independent technical and biological replicates (n, N) | Images per technical replicate (tissue) | n to calculate statistics/error bars | Statistical test |
|---|---|---|---|---|
| F5-S5F | Pax7+/MyoD- and Pax7+/MyoD+(Control 1DPE: n=7 across N=3 3DPE: n=8 across N=3 7DPE: n=9 across N=3 Y-27632 1DPE: n=9 across N=3 3DPE: n=9 across N=3 7DPE: n=9 across N=3) | 25 | Pax7+/MyoD- and Pax7+/MyoD+ (Control 1DPE: n=7 3DPE: n=8 7DPE: n=9 Y-27632 1DPE: n=9 3DPE: n=9 7DPE: n=9) | One-way ANOVA with Tukey's post-test within each individual timepoints |
| F7-S1A | Young Day 1: n=6 across N=2 Day 3: n=6 across N=2 Young + wortmannin Day 1: n=6 across N=2 Day 3: n=6 across N=2 Aged Day 1: n=6 across N=2 Day 3: n=6 across N=2 Aged + wortmannin Day 1: n=6 across N=2 Day 3: n=6 across N=2 | 104 | Young Day 1: n=6 Day 3: n=6 Young + wortmannin Day 1: n=6 Day 3: n=6 Aged Day 1: n=6 Day 3: n=6 Aged + wortmannin Day 1: n=6 Day 3: n=6 | One-way ANOVA with Tukey's post-test comparing each experimental group at the 3 DPE timepoint |
| F7-S1B | Young n=2716 across N=2 Young + wortmannin n=565 across N=2 Aged n=4437 across N=2 Aged + wortmannin n=1897 across N=2 | 104 | Young n=2716 Young + wortmannin n=565 Aged n=4437 Aged + wortmannin n=1897 | Outliers removed with the ROUT method (with Q=1%) and one-way ANOVA performed with Šidák's post-test comparing pre-selected conditions |
| F7-S2B | Young n=9 across N=3 Young + wortmannin n=6 across N=2 Aged n=8 across N=3 Aged + ortmannin n=7 across N=3 | 25 | Young n=9 Young + wortmannin n=6 Aged n=8 Aged + wortmannin n=7 | One-way ANOVA with Tukey's post-test |

FluoView V4.2b imaging software was used along with a 40× silicone immersion objective (NA 1.25; Olympus, #UPLSAPO40XS).

## Z-line analysis

Myotubes were first stained for DAPI, SAA, and actin (phalloidin) and imaged at 40× (water immersion, Two Peak autofocus, NA 1.1, and Binning of 1) using the Perkin-Elmer Opera Phenix Plus High-Content Screening System. Images were then converted to max intensity projection images using the ImageJ-BIOP Operetta Import Plugin, the channels split and saved as individual TIFF files. The SAA and actin channels were then analysed using zlineDetection on MATLAB (R2022b) (*Morris et al., 2020*). The author recommended settings were used with the following modifications:

Dot Product Threshold = 0.77
Actin Segmentation Grid Size = 150 pixels
Actin Segmentation Threshold = 0.87
Noise Removal Area = 16 pixels
Skeletonization Branch Size Removal = 5

Results for each image were then averaged per tissue.

## Bio-image analysis

For SAA coverage, stitched images were used along with a previously published ImageJ macro (*McColl et al., 2016*). The SAA signal was put in red, the threshold set to 0–45, and the tissue outline selected using the oval tool. For fusion index, cell counting, cell morphology, YFP/GFP coverage and

mean nuclear intensity, the CellProfiler software was utilized. CellProfiler version 4.2.1 (*Stirling et al., 2021*) was downloaded from source website (https://www.cellprofiler.org) and installed on a PC (Intel Core i9-11900 @ 2.5 GHz, 64.0 GB RAM, and 64-bit Windows 11 operating system). Analysis pipelines were created for each of the above-mentioned metrics. *Fusion index*: 9 × 20× max projected images were taken per tissue. The channels were split, the fiber and nuclei signal individually identified and overlayed to calculate the percentage of nuclei in fibers. *Cell counting and morphology*: 25 × 20× max projected images were taken per tissue. The channels were split, mononucleated DAPI⁺YFP⁺Pax7⁺ (or caveolin-1⁺) objects extracted using the IdentifyPrimaryObjects module and counted. For object segmentation, the global minimum cross-entropy thresholding method (*Li and Lee, 1993*) was selected. Pixel intensity and object shape were used as metrics to distinguish and segment clumped objects. Morphology measurements of the identified cellular objects were recorded using the MeasureObjectSizeShape module. For the proportion of c-FOS⁺, Ki67⁺, MyoD⁺, and/or CalcR⁺ cells, this fourth channel was overlayed over the identified objects and divided. *YFP/GFP coverage*: 25 × 20× max projected images were taken per tissue. The channels were split, the YFP/GFP signal identified, and coverage calculated using the MeasureAreaOccupied function. *Mean nuclear intensity*: 104 × 40× max projected images were taken per well. The channels were split, the nuclei, cell, and cytoplasm identified as primary, secondary, and tertiary objects, and the intensity of the FoxO3a signal within the nuclei calculated using the MeasureObjectIntensity function.

## MTS assay

To quantify the metabolic activity of myotube templates, the MTS assay was used (abcam, #ab197010). First, 200 µL of fresh DM was added to each tissue. Then, 20 µL of the MTS tetrazolium compound was added to each well and incubated for 2 hr at 37°C. The media was vigorously mixed with a pipette every 30 min to ensure maximal diffusion of the formazan dye product. The entire culture media from each tissue was then pipetted into a clear 96-well plate (Sarstedt, #83.3924) and the OD at 490 nm quantified with a spectrophotometer (Tecan, Infinite M200 Pro). The assay was performed on different tissues on different days of culture, each with their own 'media + MTS' negative control, which was subtracted as background from all OD values.

## EdU assay experiments

EdU experiments were performed using the Invitrogen Click-iT Plus EdU Alexa Fluor 555 Imaging Kit (#C10638). EdU was added to the culture media on day 5 after MuSC engraftment and refreshed every 24 hr until 7 DPE. After tissue fixation and blocking, EdU labelling was done according to the product protocol apart from a 20 min incubation instead of 30 min. Subsequent immunolabelling was done as described above. The CellProfiler pipeline was then implemented to identify mononucleated DAPI⁺YFP⁺Ki67⁻ objects and then overlayed with the EdU channel to quantify the proportion of EdU positive cells.

## Barium chloride tissue injury

On day 12 of differentiation (7 DPE), the culture media was removed, and tissues were incubated with either PSS (*Table 1*) or a 2.4 % wt/vol $BaCl_2$ solution diluted in PSS for a period of 4 hr (protocol adapted from previously published literature; *Morton et al., 2019*). Tissues were then washed 3×5 min with warm wash media (*Table 1*) and then returned to fresh DM for 2 more days before fixation.

## 2D culture experiments

For 2D myotube culture experiments, microwells were first coated with a 5% v/v Geltrex/DMEM solution for 1 hr at 37°C. After drying, 40,000 primary myoblasts were added per well in 200 µL of GM, which was then switched to DM after 2 days. The cell density was selected to match that observed in 3D myotube templates, which is ≈1273 cells/mm². On day 5 of differentiation, 500 MuSCs were engrafted onto 2D myotubes in a 4 µL volume.

## Statistical analysis

All data generated and analysed during this study are included in the manuscript files. Statistical analysis was performed using the GraphPad Prism 9 software. Most experiments were performed with three technical tissue replicates per experimental group and repeated on three independent

occasions (i.e., n=9 technical replicates across N=3 biological replicates). Please refer to *Table 3* for a specific breakdown of replicates per experiment and for documentation of ROUT outlier statistical tests justifying removal of any data points. Source Data files containing all of the numerical data used to generate each of the figures have been provided. All error bars show standard error of the mean (s.e.m.). Significance was defined as $p \leq 0.05$.

## Acknowledgements

This project was funded by a CIHR CGS-D scholarship to EJ, an Ontario Graduate Scholarship to EJ, a Cecil Yip Doctoral Award to EJ, a MITACS Globalink Research Award to EJ, a Michael Smith Foreign Study Supplement to EJ, a Medicine by Design Canada First Research Excellence Fund grant to PMG (MbDC2-2019-02), and a Canada Research Chair in Endogenous Repair to PMG. We thank Dr Christophe Vanbelle from la Plateforme d'Imagerie Cellulaire du Centre de Recherche en Cancérologie de Lyon for enabling timely revision experiments and Dr. Louise Moyle for cryopreserving skeletal muscle tissues from the aged mice used in this study. We also thank Olivier Burri and the BioImaging and Optics Core Facility from l'École Polytechnique Fédérale de Lausanne for their help using BIOP-Fiji plugins for data analysis.

## Additional information

### Funding

| Funder | Grant reference number | Author |
|---|---|---|
| Canadian Institutes of Health Research | CGS-D Scholarship | Erik Jacques |
| Ontario Ministry of Research and Innovation | Ontario Graduate Scholarship | Erik Jacques |
| Mitacs | Globalink Research Award | Erik Jacques |
| Canadian Institutes of Health Research | Michael Smith Foreign Study Supplement | Erik Jacques |
| Canada First Research Excellence Fund | Medicine by Design (MbDC2-2019-02) | Penney M Gilbert |
| Natural Sciences and Engineering Research Council of Canada | Canada Research Chair in Endogenous Repair | Penney M Gilbert |

The funders had no role in study design, data collection and interpretation, or the decision to submit the work for publication.

### Author contributions

Erik Jacques, Conceptualization, Data curation, Software, Formal analysis, Supervision, Funding acquisition, Investigation, Visualization, Methodology, Writing - original draft, Project administration, Writing - review and editing; Yinni Kuang, Software, Formal analysis, Validation, Investigation, Visualization, Methodology, Writing - original draft, Writing - review and editing; Allison P Kann, Robert S Krauss, Resources, Writing - review and editing; Fabien Le Grand, Supervision, Funding acquisition, Writing - review and editing; Penney M Gilbert, Conceptualization, Data curation, Supervision, Funding acquisition, Writing - original draft, Project administration, Writing - review and editing

### Author ORCIDs

Allison P Kann (iD) http://orcid.org/0000-0003-0111-9081
Fabien Le Grand (iD) http://orcid.org/0000-0002-7843-3899
Robert S Krauss (iD) http://orcid.org/0000-0002-7661-3335
Penney M Gilbert (iD) http://orcid.org/0000-0001-5509-9616

## Ethics

All animal use protocols were reviewed and approved by the local Animal Care Committee (ACC) within the Division of Comparative Medicine (DCM) at the University of Toronto. All methods in this study were conducted as described in the approved animal use protocols (#20012838) and more broadly in accordance with the guidelines and regulations of the DCM ACC and the Canadian Council on Animal Care.

## Decision letter and Author response

Decision letter https://doi.org/10.7554/eLife.81738.sa1
Author response https://doi.org/10.7554/eLife.81738.sa2

## Additional files

### Supplementary files
• MDAR checklist

### Data availability
All data generated and analysed during this study are included in the manuscript files. In addition, a source data file containing all of the numerical data used to generate each of the figures has been provided.

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
