## [Editor Report]

This methods paper explores procedures evaluating the balance between muscle cell quiescence and activation. These could well permit investigations of long-standing questions in key areas of muscle function. The latter include the regulation of adult stem cell pool size and functional heterogeneities in this, as well as regulators of muscle quiescence.

---

## [Decision Letter]

**Decision letter after peer review:**

Thank you for submitting your article "Rescue of aged muscle stem cell quiescence defects by AKT inhibition revealed with a 3D biomimetic culture assay" for consideration by *eLife*. Your article has been reviewed by 2 peer reviewers, one of whom is a member of our Board of Reviewing Editors, and the evaluation has been overseen by a Reviewing Editor and Mone Zaidi as the Senior Editor. The following individual involved in review of your submission has agreed to reveal their identity: Colin Crist (Reviewer #2).

The reviewers have discussed their reviews with one another, and the Reviewing Editor has drafted this to help you prepare a revised submission. We include the following broad feedback in addition to the specific revisions requested below:

1) This is a timely contribution that demonstrates the utility of a myotube sheet biomimetic niche, in a 96 well format, to recapture some biological aspects of MuSC quiescence ex vivo. The format should provide a novel biological context within which features of MuSC quiescence can be studied.

2) The presented data does not fully capture the innovative 3D bioengineered myotube substrate, nor does it capture the advance that has been made, beyond more traditional 2D culture techniques. (The work is innovative, but the innovation is not captured by the presented data).

3) The title of the manuscript does not reflect the balance of the article. Although the authors have presented some statistically significant data showing Akt inhibition (Wortmannin) rescues aged MuSC defects in their assay, I would expect more lines of evidence, including rescue experiments in vivo, to support the authors conclusion that their assay revealed Akt inhibition as a targetable pathway to rescue aged MuSC quiescence.

4) The authors have innovated a novel in vitro assay that should be useful to assess more broadly critical MuSC stem cell properties, which would have great appeal, but the authors have somewhat restricted their analyses and interpretations to study quiescence. Experimentally, the authors could address more broadly stem cell properties including donor cell contribution to self-renewal/quiescence (as they have done) and differentiation (YFP/MYOD; YFP/myogenin; YFP/MHC). Ideally, if the 3D culture conditions maintain an adult stem cell population, mononuclear YFP cells should be reisolated and engrafted in vitro a second time, followed by their reassessment of their stem cell properties.

Essential revisions:

1. The data presented in Figure 1 does not capture the innovation in the work. The novel aspects of the Figure that I can appreciate are the method presented in Figure 1A (but please indicate seeding density) and the timecourse of stable myotubes present after 18 days of culture in a 96-well format, which to my knowledge cannot be achieved in 2D culture. I am unable to appreciate by the presented data the nature of the 3D myotube sheets, compared to conventional culture methods.

Suggestions: (1) present higher magnification, higher resolution images to capture greater maturity, organization and/or structural features of 3D myotube sheets. (2) Present data with direct comparison to 2D myotube cultures. Some comparison has been achieved in Figure 4, but this is a different analysis of the seeding.

2. In Figure 2, the authors provide evidence that varying numbers of donor, YFP(+), MuSCs contribute an ex vivo MuSC compartment, and quantify the numbers of YFP(+) PAX7(+) MuSCs at various timepoints after 'engraftment' (Figure 2B-D). It is not clear what happens to YFP(+) donor cells that no longer express PAX7. Do they contribute to new or existing myotubes as suggested by the illustration (Figure 2A)?

3. Figure 4 of the paper, which compares different conditions that enable persistent MuSC population in vitro, is amongst the most important Figures in the paper. In this figure, the reader can appreciate the innovation of the 3D myotube system (Figure 3C), compared to other conditions, for example 2D myotubes (Figure 3F) and ECM/cellulose only (Figure 3E). To have greater impact, I suggest a deeper analysis as has been done in other Figures, for example, quantification of PAX7, nuclear morphology, cytoplasmic projections, lack of MYOD and Ki67 immunolabeling. Please also include representative images.

4. Data presented in Figure 6 showing MuSC niche interactions is very interesting, and I appreciate the authors are pushing technology boundaries, but the description of the interactions as 'an uncommon occurrence' is problematic and it does not indicate which of the data is uncommon (N-cadherin tips or integrin/cadherin immunolabeling, or both?) I suggest quantifying the frequency of each of the observations reported in Figure 6 at minimum.

5. Figure 7A, F, G. Specifically with respect to wortmannin treatment, the authors have presented some data that is statistically significant and some data that is not statistically significant. More importantly, the biological significance of the reported data does not always support the authors conclusions.

6. If the authors want to conclude that their innovative in vitro 3D niche system led to the identification of wortmannin to rescue aged MuSC quiescence defects, greater supporting evidence is required, including critical in vivo administration of wortmannin to aging mice and a deeper analysis of rescued MuSC properties/regenerative capacity. Such an experimental revision is not necessarily required and may be beyond the scope of the work, but at minimum the authors need to reassess their conclusions and title of the manuscript.

---

## [Author Response]

The reviewers have discussed their reviews with one another, and the Reviewing Editor has drafted this to help you prepare a revised submission.Essential revisions:1. The data presented in Figure 1 does not capture the innovation in the work. The novel aspects of the Figure that I can appreciate are the method presented in Figure 1A (but please indicate seeding density) and the timecourse of stable myotubes present after 18 days of culture in a 96-well format, which to my knowledge cannot be achieved in 2D culture. I am unable to appreciate by the presented data the nature of the 3D myotube sheets, compared to conventional culture methods.Suggestions: (1) present higher magnification, higher resolution images to capture greater maturity, organization and/or structural features of 3D myotube sheets. (2) Present data with direct comparison to 2D myotube cultures. Some comparison has been achieved in Figure 4, but this is a different analysis of the seeding.

With Figure 1, we aimed to identify (a) the time-point when myotube formation had plateaued, as this would inform when we would initially add the MuSCs and (b) the “culture window” wherein we observed morphologically sound and metabolically active myotubes, as quiescence assay end-points should occur before tissue degradation owing to the confounding MuSC ‘activation’ that would be induced. We now make these goals clear in the text.

As it is established in the field that myotubes cannot be maintained in 2D culture for prolonged periods due to eventual substrate detachment, we had focused our analysis on the outcomes of the MuSCs in each condition rather than the myotubes themselves in Figure 4. However, we agree that an additional layer of insight is offered by highlighting differences between myotubes formed in our 3D model, when compared to 2D myotubes. We thus added new Figure 1—figure supplement 2 comparing myotubes cultivated in 2D as compared to 3D, showing both stitched and high-resolution images, which addresses differences in maturity by analyzing the striations through sarcomeric α-actinin staining. We’ve also updated the Results accordingly (lines 162-179).

2. In Figure 2, the authors provide evidence that varying numbers of donor, YFP(+), MuSCs contribute an ex vivo MuSC compartment, and quantify the numbers of YFP(+) PAX7(+) MuSCs at various timepoints after 'engraftment' (Figure 2B-D). It is not clear what happens to YFP(+) donor cells that no longer express PAX7. Do they contribute to new or existing myotubes as suggested by the illustration (Figure 2A)?

Thank you for this comment, which we believe we addressed in the initial submission through our experiments in Figure 3 and Figure 3—figure supplement 3, and as discussed in lines 250-254 and 254-266. We can see that the engrafted YFP+ Pax7+ MuSCs remain a stable population size over the 7 days (Figure 2D). At 3 DPE we note transient increase in Pax7+Ki67+ (Figure 3C), Pax7+MyoD+ (Figure 5—figure supplement 3A, Figure 5—figure supplement 4D), and Pax7- cells (Figure below, left (n=9 across N=3 biological replicates)). By 7 DPE these population proportions have reduced, and we see an accumulation YFP+ myotubes (Figure 3—figure supplement 3B), however we see no EdU+ myonuclei. From these pieces of evidence, we then concluded that some cells were directly fusing (with one another to form de novo myotubes, or with existing myotubes) while others continue to proliferate while maintaining Pax7 expression. There is yet another population of mono-nucleated Pax7- cells remaining at 7 DPE that do not appear to be MyoD+ and we speculate they are stuck in a state of terminal differentiation.

To recap, we’ve put together an illustration of our current model for explaining engrafted donor cell fate (Author response image 1, right). Starting off with activated MuSCs that are engrafted, the majority immediately inactivate. Other, perhaps those too far down the activation process, directly differentiate and fuse to existing myotubes. A portion also proliferates, but again primarily exit cell-cycle and inactivate downstream, we theorize, to replace the cells lost to fusion.

**Author response image 1. sa2fig1:** 

3. Figure 4 of the paper, which compares different conditions that enable persistent MuSC population in vitro, is amongst the most important Figures in the paper. In this figure, the reader can appreciate the innovation of the 3D myotube system (Figure 3C), compared to other conditions, for example 2D myotubes (Figure 3F) and ECM/cellulose only (Figure 3E). To have greater impact, I suggest a deeper analysis as has been done in other Figures, for example, quantification of PAX7, nuclear morphology, cytoplasmic projections, lack of MYOD and Ki67 immunolabeling. Please also include representative images.

We agree with the reviewer on the importance of this Figure and thank them for this suggestion. Along with the population kinetics and Ki67 labelling and analysis over time that was included in the first submission, we’ve now added representative images showing the different abundance of Pax7+ and Ki67+ donor cells (Figure 4G and lines 289-290). As well, of the three conditions with a non-negligible amount of Pax7+ donor cells remaining at 7 DPE (2D, 3D myotubes, and 2D myotubes), we’ve now analyzed the morphology by reporting the max\min feret diameter ratio and nuclear eccentricity of those cells (Figure 5—figure supplement 1 and lines 296-298 and 309-312).

4. Data presented in Figure 6 showing MuSC niche interactions is very interesting, and I appreciate the authors are pushing technology boundaries, but the description of the interactions as 'an uncommon occurrence' is problematic and it does not indicate which of the data is uncommon (N-cadherin tips or integrin/cadherin immunolabeling, or both?) I suggest quantifying the frequency of each of the observations reported in Figure 6 at minimum.

We completely agree with the reviewer on this point. For the revised submission, we now quantify and report the incidence of each phenotypic observation made in Figure 6C-D (N-cad+ projections, polarized Iα7/M-cad expression) within the mononuclear donor cell population at 7 DPE. The changes are also reflected in the Results (lines 383-384 and 386-388).

5. Figure 7A, F, G. Specifically with respect to wortmannin treatment, the authors have presented some data that is statistically significant and some data that is not statistically significant. More importantly, the biological significance of the reported data does not always support the authors conclusions.

We have edited the section heading on line 393 to “Aged MuSCs exhibit delayed inactivation in mini-IDLE that is rescuable by Akt inhibition” in order to better reflect our results, and thank the reviewer for pointing out this overstatement on our part. We have also changed the title of our manuscript to focus more so on our model, and less on the outcomes of Figure 7.

With regards to Figure 7E-G, recent work by our collaborators (Kann et al. 2022 *Cell Stem Cell,* An injury-responsive Rac-to-Rho GTPase switch drives activation of muscle stem cells through rapid cytoskeletal remodeling) showed that the morphology of the MuSC reflects its activation state. We’ve also shown several lines of evidence in the context of our model that align with their findings (Figure 5D-F, Figure 5—figure supplement 1, Figure 5—figure supplement 3, Figure 5—figure supplement 4). Hence, with our image analysis in our Young vs Aged studies, we sought to understand whether the aged MuSCs were attaining the same “quiescent-like morphology” we observed with young MuSCs. The aged MuSCs did not, and resembled a more activated morphology (rounded). Treatment with Wortmannin did not have a pronounced effect on this front, aside from a subtle change in nuclear eccentricity (0.75 to 0.81) which was visibly different (see 0.06 eccentricity difference example, Author response image 2). Since changes to nuclear morphology can modulate gene expression, we felt it necessary to report in the main figure. Our findings may also help inform future studies by others as well, which justifies in our mind the inclusion of negative data. We have modified the text to tone down conclusions drawn from this study.

6. If the authors want to conclude that their innovative in vitro 3D niche system led to the identification of wortmannin to rescue aged MuSC quiescence defects, greater supporting evidence is required, including critical in vivo administration of wortmannin to aging mice and a deeper analysis of rescued MuSC properties/regenerative capacity. Such an experimental revision is not necessarily required and may be beyond the scope of the work, but at minimum the authors need to reassess their conclusions and title of the manuscript.

We have changed the title of our manuscript to “mini-IDLE: A 3D biomimetic culture assay to interrogate mechanisms governing muscle stem cell quiescence and niche repopulation”, to focus more so on our model and we also adjusted our conclusion statements (lines 450-451 and line 462).